# CM3: Cooperative Multi-goal Multi-stage Multi-agent Reinforcement Learning

**Jiachen Yang**[*1], **Alireza Nakhaei**[†3], **David Isele**[2], **Kikuo Fujimura**[2] **& Hongyuan Zha**[1]
[1]Georgia Institute of Technology
[2]Honda Research Institute
[3]Toyota Research Institute

## Abstract

A variety of cooperative multi-agent control problems require agents to achieve individual goals while contributing to collective success. This multi-goal multi-agent setting poses difficulties for recent algorithms, which primarily target settings with a single global reward, due to two new challenges: efficient exploration for learning both individual goal attainment and cooperation for others' success, and credit-assignment for interactions between actions and goals of different agents. To address both challenges, we restructure the problem into a novel two-stage curriculum, in which single-agent goal attainment is learned prior to learning multi-agent cooperation, and we derive a new multi-goal multi-agent policy gradient with a credit function for localized credit assignment. We use a function augmentation scheme to bridge value and policy functions across the curriculum. The complete architecture, called CM3, learns significantly faster than direct adaptations of existing algorithms on three challenging multi-goal multi-agent problems: cooperative navigation in difficult formations, negotiating multi-vehicle lane changes in the SUMO traffic simulator, and strategic cooperation in a Checkers environment.

## 1 Introduction

Many real-world scenarios that require cooperation among multiple autonomous agents are *multi-goal* multi-agent control problems: each agent needs to achieve its own individual goal, but the global optimum where all agents succeed is only attained when agents cooperate to allow the success of other agents. In autonomous driving, multiple vehicles must execute cooperative maneuvers when their individual goal locations and nominal trajectories are in conflict (e.g., double lane merges) (Cao *et al.*, 2013). In social dilemmas, mutual cooperation has higher global payoff but agents' individual goals may lead to defection out of fear or greed (Van Lange *et al.*, 2013). Even settings with a global objective that seem unfactorizable can be formulated as multi-goal problems: in Starcraft II micromanagement, a unit that gathers resources must not accidentally jeopardize a teammate's attempt to scout the opponent base (Blizzard Entertainment, 2019); in traffic flow optimization, different intersection controllers may have local throughput goals but must cooperate for high global performance (Zhang *et al.*, 2019). While the framework of multi-agent reinforcement learning (MARL) (Littman, 1994; Stone and Veloso, 2000; Shoham *et al.*, 2003) has been equipped with methods in deep reinforcement learning (RL) (Mnih *et al.*, 2015; Lillicrap *et al.*, 2016) and shown promise on high-dimensional problems with complex agent interactions (Lowe *et al.*, 2017; Mordatch and Abbeel, 2018; Foerster *et al.*, 2018; Lin *et al.*, 2018; Srinivasan *et al.*, 2018), learning multi-agent cooperation in the multi-goal scenario involves significant open challenges.

First, given that exploration is crucial for RL (Thrun, 1992) and even more so in MARL with larger state and joint action spaces, how should agents explore to learn both individual goal attainment and cooperation for others' success? Uniform random exploration is common in deep MARL (Hernandez-Leal *et al.*, 2018) but can be highly inefficient as the value of cooperative actions may be discoverable only in small regions of state space where cooperation is needed. Furthermore, the conceptual difference between attaining one's own goal and cooperating for others' success calls for

---

[*]Correspondence to jiachen.yang@gatech.edu
[†]Work done at HRI

more modularized and targeted approaches. Second, while there are methods for multi-agent credit assignment when all agents share a single goal (i.e., a global reward) (Chang *et al.*, 2004; Foerster *et al.*, 2018; Nguyen *et al.*, 2018), and while one could treat the cooperative *multi-goal* scenario as a problem with a single joint goal, this coarse approach makes it extremely difficult to evaluate the impact of an agent's action on another agent's success. Instead, the multi-goal scenario can benefit from fine-grained credit assignment that leverages available structure in action-goal interactions, such as local interactions where only few agents affect another agent's goal attainment at any time.

Given these open challenges, our paper focuses on the cooperative multi-goal multi-agent setting where each agent is assigned a goal[1] and must learn to cooperate with other agents with possibly different goals. To tackle the problems of efficient exploration and credit assignment in this complex problem setting, we develop CM3, a novel general framework involving three synergistic components:

1. We approach the difficulty of multi-agent exploration from a novel curriculum learning perspective, by first training an actor-critic pair to achieve different goals in an *induced single-agent setting* (Stage 1), then using them to initialize all agents in the multi-agent environment (Stage 2). The key insight is that agents who can already act toward individual objectives are better prepared for discovery of cooperative solutions with additional exploration once other agents are introduced. In contrast to hierarchical learning where sub-goals are selected sequentially in time (Sutton *et al.*, 1999), all agents act toward their goals simultaneously in Stage 2 of our curriculum.

2. Observing that a wide array of complex MARL problems permit a decomposition of agents' observations and state vectors into components of self, others, and non-agent specific environment information (Hernandez-Leal *et al.*, 2018), we employ function augmentation to bridge Stages 1-2: we reduce the number of trainable parameters of the actor-critic in Stage 1 by limiting their input space to the part that is sufficient for single-agent training, then augment the architecture in Stage 2 with additional inputs and trainable parameters for learning in the multi-agent environment.

3. We propose a *credit function*, which is an action-value function that specifically evaluates action-goal pairs, for localized credit assignment in multi-goal MARL. We use it to derive a multi-goal multi-agent policy gradient for Stage 2. In synergy with the curriculum, the credit function is constructed via function augmentation from the critic in Stage 1.

We evaluate our method on challenging multi-goal multi-agent environments with high-dimensional state spaces: cooperative navigation with difficult formations, double lane merges in the SUMO simulator (Lopez *et al.*, 2018), and strategic teamwork in a Checkers game. CM3 solved all domains significantly faster than IAC and COMA (Tan, 1993; Foerster *et al.*, 2018), and solved four out of five environments significantly faster than QMIX (Rashid *et al.*, 2018). Exhaustive ablation experiments show that the combination of all three components is crucial for CM3's overall high performance.

## 2 RELATED WORK

While early theoretical work analyzed Markov games in discrete state and action spaces (Tan, 1993; Littman, 1994; Hu and Wellman, 2003), recent literature have leveraged techniques from deep RL to develop general algorithms for high dimensional environments with complex agent interactions (Tampuu *et al.*, 2017; Mordatch and Abbeel, 2018; Lowe *et al.*, 2017), which pose difficulty for traditional methods that do not generalize by learning interactions (Bhattacharya *et al.*, 2010).

Cooperative multi-agent learning is important since many real-world problems can be formulated as distributed systems in which decentralized agents must coordinate to achieve shared objectives (Panait and Luke, 2005). The multi-agent credit assignment problem arises when agents share a global reward (Chang *et al.*, 2004). While credit assignment be resolved when independent individual rewards are available (Singh *et al.*, 2019), this may not be suitable for the fully cooperative setting: Austerweil *et al.* (2016) showed that agents whose rewards depend on the success of other agents can cooperate better than agents who optimize for their own success. In the special case when all agents have a single goal and share a global reward, COMA (Foerster *et al.*, 2018) uses a counterfactual baseline, while Nguyen *et al.* (2018) employs count-based variance reduction limited to discrete-state

---

[1]Goal discovery and assignment are challenges for MARL. However, many practical multi-agent problems have clear goal assignments, such as in autonomous driving and soccer. Our work is specific to known goal assignment and is complementary to methods such as (Carion *et al.*, 2019) for the unknown case.

environments. However, their centralized critic does not evaluate the specific impact of an agent's action on another's success in the general multi-goal setting. When a global objective is the sum of agents' individual objectives, value-decomposition methods optimize a centralized Q-function while preserving scalable decentralized execution (Sunehag *et al.*, 2018; Rashid *et al.*, 2018; Son *et al.*, 2019), but do not address credit assignment. While MADDPG (Lowe *et al.*, 2017) and M3DDPG (Li *et al.*, 2019) apply to agents with different rewards, they do not address multi-goal cooperation as they do not distinguish between cooperation and competition, despite the fundamental difference.

Multi-goal MARL was considered in Zhang *et al.* (2018), who analyzed convergence in a special networked setting restricted to fully-decentralized training, while we conduct centralized training with decentralized execution (Oliehoek *et al.*, 2008). In contrast to multi-*task* MARL, which aims for generalization among *non-simultaneous* tasks (Omidshafiei *et al.*, 2017), and in contrast to hierarchical methods that *sequentially* select subtasks (Vezhnevets *et al.*, 2017; Shu and Tian, 2019), our decentralized agents must cooperate *concurrently* to attain all goals. Methods for optimizing high-level agent-task assignment policies in a hierarchical framework (Carion *et al.*, 2019) are complementary to our work, as we focus on learning low-level cooperation after goals are assigned. Prior application of curriculum learning (Bengio *et al.*, 2009) to MARL include a single cooperative task defined by the number of agents (Gupta *et al.*, 2017) and the probability of agent appearance (Sukhbaatar *et al.*, 2016), without explicit individual goals. Rusu *et al.* (2016) instantiate new neural network columns for task transfer in single-agent RL. Techniques in transfer learning (Pan and Yang, 2010) are complementary to our novel curriculum approach to MARL.

## 3 PRELIMINARIES

In multi-goal MARL, each agent should achieve a goal drawn from a finite set, cooperate with other agents for collective success, and act independently with limited local observations. We formalize the problem as an episodic multi-goal Markov game, review an actor-critic approach to centralized training of decentralized policies, and summarize counterfactual-based multi-agent credit assignment.

**Multi-goal Markov games.** A multi-goal Markov game is a tuple $\langle \mathcal{S}, \{\mathcal{O}^n\}, \{\mathcal{A}^n\}, P, R, \mathcal{G}, N, \gamma \rangle$ with $N$ agents labeled by $n \in [N]$. In each episode, each agent $n$ has one fixed goal $g^n \in \mathcal{G}$ that is known only to itself. At time $t$ and global state $s_t \in \mathcal{S}$, each agent $n$ receives an observation $o_t^n := o^n(s_t) \in \mathcal{O}^n$ and chooses an action $a_t^n \in \mathcal{A}^n$. The environment moves to $s_{t+1}$ due to joint action $\mathbf{a}_t := \{a_t^1, \dots, a_t^N\}$, according to transition probability $P(s_{t+1}|s_t, \mathbf{a}_t)$. Each agent receives a reward $R_t^n := R(s_t, \mathbf{a}_t, g^n)$, and the learning task is to find stochastic decentralized policies $\pi^n \colon \mathcal{O}^n \times \mathcal{G} \times \mathcal{A}^n \to [0, 1]$, conditioned only on local observations and goals, to maximize $J(\boldsymbol{\pi}) := \mathbb{E}_{\boldsymbol{\pi}} \left[ \sum_{t=0}^{\infty} \gamma^t \sum_{n=1}^{N} R(s_t, \mathbf{a}_t, g^n) \right]$, where $\gamma \in (0, 1)$ and joint policy $\boldsymbol{\pi}$ factorizes as $\boldsymbol{\pi}(\mathbf{a}|s, \mathbf{g}) := \prod_{n=1}^{N} \pi^n(a^n|o^n, g^n)$ due to decentralization. Let $a^{-n}$ and $g^{-n}$ denote all agents' actions and goals, respectively, *except* that of agent $n$. Let boldface $\mathbf{a}$ and $\mathbf{g}$ denote the joint action and joint goals, respectively. For brevity, let $\pi(a^n) := \pi^n(a^n|o^n, g^n)$. This model covers a diverse set of cooperation problems in the literature (Hernandez-Leal *et al.*, 2018), without constraining how the attainability of a goal depends on other agents: at a traffic intersection, each vehicle can easily reach its target location if not for the presence of other vehicles; in contrast, agents in a strategic game may not be able to maximize their rewards in the absence of cooperators (Sunehag *et al.*, 2018).

**Centralized learning of decentralized policies.** A centralized critic that receives full state-action information can speed up training of decentralized actors that receive only local information (Lowe *et al.*, 2017; Foerster *et al.*, 2018). Directly extending the single-goal case, for each $n \in [1..N]$ in a multi-goal Markov game, critics are represented by the value function $V_n^{\boldsymbol{\pi}}(s) := \mathbb{E}_{\boldsymbol{\pi}} \left[ \sum_{t=0}^{\infty} \gamma^t R_t^n \mid s_0 = s \right]$ and the action-value function $Q_n^{\boldsymbol{\pi}}(s, \mathbf{a}) := \mathbb{E}_{\boldsymbol{\pi}} \left[ \sum_{t=0}^{\infty} \gamma^t R_t^n \mid s_0 = s, \mathbf{a}_0 = \mathbf{a} \right]$, which evaluate the joint policy $\boldsymbol{\pi}$ against the reward $R^n$ for each goal $g^n$.

**Multi-agent credit assignment.** In MARL with a single team objective, COMA addresses credit assignment by using a counterfactual baseline in an advantage function $A^n(s, \mathbf{a}) := Q^{\boldsymbol{\pi}}(s, \mathbf{a}) - \sum_{\hat{a}^n} \pi^n(\hat{a}^n|o^n)Q^{\boldsymbol{\pi}}(s, (\hat{a}^n, a^{-n}))$ (Foerster *et al.*, 2018, Lemma 1), which evaluates the contribution of a chosen action $a^n$ versus the average of all possible counterfactuals $\hat{a}^n$, keeping $a^{-n}$ fixed. The analysis in Wu *et al.* (2018) for a formally equivalent action-dependent baseline in RL suggests that COMA is a low-variance estimator for single-goal MARL. We derive its variance in Appendix C.1. However, COMA is unsuitable for credit assignment in multi-goal MARL, as it would treat the

collection of goals **g** as a global goal and only learn from total reward, making it extremely difficult to disentangle each agent's impact on other agents' goal attainment. Furthermore, a global Q-function does not explicitly capture structure in agents' interactions, such as local interactions involving a limited number of agents. We substantiate these arguments by experimental results in Section 6.

## 4 METHODS

We describe the complete CM3 learning framework as follows. First we define a credit function as a mechanism for credit assignment in multi-goal MARL, then derive a new cooperative multi-goal policy gradient with localized credit assignment. Next we motivate the possibility of significant training speedup via a curriculum for multi-goal MARL. We describe function augmentation as a mechanism for efficiently bridging policy and value functions across the curriculum stages, and finally synthesize all three components into a synergistic learning framework.

### 4.1 CREDIT ASSIGNMENT IN MULTI-GOAL MARL

If all agents take greedy goal-directed actions that are individually optimal in the absence of other agents, the joint action can be sub-optimal (e.g. straight-line trajectory towards target in traffic). Instead rewarding agents for both individual and collective success can avoid such bad local optima. A naïve approach based on previous works (Foerster *et al.*, 2018; Lowe *et al.*, 2017) would evaluate the joint action **a** via a global Q-function $Q_n^{\boldsymbol{\pi}}(s, \mathbf{a})$ for each agent's goal $g^n$, but this does not precisely capture each agent's contribution to another agent's attainment of its goal. Instead, we propose an explicit mechanism for credit assignment by learning an additional function $Q_n^{\boldsymbol{\pi}}(s, a^m)$ that evaluates pairs of action $a^m$ and goal $g^n$, for use in a multi-goal actor-critic algorithm. We define this function and show that it satisfies the classical relation needed for sample-based model-free learning.

**Definition 1.** For $n, m \in [N]$, $s \in \mathcal{S}$, the *credit function* for goal $g^n$ and $a^m \in \mathcal{A}^m$ by agent $m$ is:

$$Q_n^{\boldsymbol{\pi}}(s, a^m) := \mathbb{E}_{\boldsymbol{\pi}}\Big[\sum_{t=0}^{\infty} \gamma^t R_t^n \mid s_0 = s, a_0^m = a^m\Big] \tag{1}$$

**Proposition 1.** *For all $m, n \in [N]$, the credit function* (1) *satisfies the following relations:*

$$Q_n^{\boldsymbol{\pi}}(s, a^m) = \mathbb{E}_{\boldsymbol{\pi}}\big[R_t^n + \gamma Q_n^{\boldsymbol{\pi}}(s_{t+1}, a_{t+1}^m) \mid s_t = s, a_t^m = a^m\big] \tag{2}$$

$$V_n^{\boldsymbol{\pi}}(s) = \sum_{a^m} \pi^m(a^m|o^m, g^m) Q_n^{\boldsymbol{\pi}}(s, a^m) \tag{3}$$

Derivations are given in Appendix B.1, including the relation between $Q_n^{\boldsymbol{\pi}}(s, a^m)$ and $Q_n^{\boldsymbol{\pi}}(s, \mathbf{a})$. Equation (2) takes the form of the Bellman expectation equation, which justifies learning the credit function, parameterized by $\theta_{Q_c}$, by optimizing the standard loss function in deep RL:

$$L(\theta_{Q_c}) = \mathbb{E}_{\boldsymbol{\pi}}\Big[\big(R_t^n + \gamma Q_n^{\boldsymbol{\pi}}(s_{t+1}, a_{t+1}^m; \theta_{Q_c}) - Q_n^{\boldsymbol{\pi}}(s_t, a_t^m; \theta_{Q_c})\big)^2\Big] \tag{4}$$

While centralized training means the input space scales linearly with agent count, many practical environments involving only *local* interactions between agents allows centralized training with few agents while retaining decentralized performance when deployed at scale (evidenced in Appendix E).

### 4.2 COOPERATIVE MULTI-GOAL MULTI-AGENT POLICY GRADIENT

We use the credit function as a critic within a policy gradient for multi-goal MARL. Letting $\theta$ parameterize $\boldsymbol{\pi}$, the overall objective $J(\boldsymbol{\pi})$ is maximized by ascending the following gradient:

**Proposition 2.** *The cooperative multi-goal credit function based MARL policy gradient is*

$$\nabla_\theta J(\boldsymbol{\pi}) = \mathbb{E}_{\boldsymbol{\pi}}\Big[\sum_{m,n=1}^{N} \big(\nabla_\theta \log \pi^m(a^m|o^m, g^m)\big) A_{n,m}^{\boldsymbol{\pi}}(s, \mathbf{a})\Big] \tag{5}$$

$$A_{n,m}^{\boldsymbol{\pi}}(s, \mathbf{a}) := Q_n^{\boldsymbol{\pi}}(s, \mathbf{a}) - \sum_{\hat{a}^m} \pi^m(\hat{a}^m|o^m, g^m) Q_n^{\boldsymbol{\pi}}(s, \hat{a}^m) \tag{6}$$

This is derived in Appendix B.2. For a fixed agent $m$, the inner summation over $n$ considers all agents' goals $g^n$ and updates $m$'s policy based on the advantage of $a^m$ over all counterfactual actions $\hat{a}^m$, as measured by the credit function for $g^n$. The strength of interaction between action-goal pairs is captured by the extent to which $Q_n^{\boldsymbol{\pi}}(s, \hat{a}^m)$ varies with $\hat{a}^m$, which directly impacts the magnitude of the gradient on agent $m$'s policy. For example, strong interaction results in non-constant $Q_n^{\boldsymbol{\pi}}(s, \cdot)$, which implies larger magnitude of $A_{n,m}^{\boldsymbol{\pi}}$ and larger weight on $\nabla_\theta \log \pi(a^m)$. The double summation accounts for first-order interaction between all action-goal pairs, but complexity can be reduced by omitting terms when interactions are known to be sparse, and our empirical runtimes are on par with other methods due to efficient batch computation (Appendix F). As the second term in $A_{n,m}^{\boldsymbol{\pi}}$ is a baseline, the reduction of variance can be analyzed similarly to that for COMA, given in Appendix C.2. While $A_{n,m}^{\boldsymbol{\pi}} = Q_n^{\boldsymbol{\pi}}(s, \mathbf{a}) - V_n^{\boldsymbol{\pi}}(s)$ (due to (3)), ablation results show stability improvement due to the credit function (Section 6). As the credit function takes in a single agent's action, it synergizes with both CM3's curriculum and function augmentation as described in Section 4.5.

## 4.3 Curriculum for multi-goal MARL

Multi-goal MARL poses a significant challenge for exploration. Random exploration can be highly inefficient for concurrently learning both individual task completion and cooperative behavior. Agents who cannot make progress toward individual goals may rarely encounter the region of state space where cooperation is needed, rendering any exploration useless for learning cooperative behavior. On the other extreme, exploratory actions taken in situations that require precise coordination can easily lead to penalties that cause agents to avoid the coordination problem and fail to achieve individual goals. Instead, we hypothesize and confirm in experiments that agents who can achieve individual goals in the absence of other agents can more reliably produce state configurations where cooperative solutions are easily discovered with additional exploration in the multi-agent environment[2].

We propose a MARL curriculum that first solves a single-agent Markov decision process (MDP), as preparation for subsequent exploration speedup. Given a cooperative multi-goal Markov game **MG**, we induce an MDP **M** to be the tuple $\langle \mathcal{S}^n, \mathcal{O}^n, A^n, P^n, R, \gamma \rangle$, where an agent $n$ is selected to be the single agent in **M**. Entities $\mathcal{S}^n$, $P^n$, and $R$ are defined by removing all dependencies on agent interactions, so that only components depending on agent $n$ remain. This reduction to **M** is possible in almost all fully cooperative multi-agent environments used in a large body of work[3] (Hernandez-Leal *et al.*, 2018), precisely because they support a variable number of agents, including $N = 1$. Important real-world settings that allow this reduction include autonomous driving, multi traffic light control, and warehouse commissioning (removing all but one car/controller/robot, respectively, from the environment). Given a full Markov game implementation, the reduction involves only deletion of components associated with all other agents from state vectors (since an agent is uniquely defined by its attributes), deletion of if-else conditions from the reward function corresponding to agent interactions, and likewise from the transition function if a simulation is used. Appendix G provides practical guidelines for the reduction. Based on **M**, we define a *greedy policy* for **MG**.

**Definition 2.** A *greedy policy* $\pi^n$ by agent $n$ for cooperative multi-goal **MG** is defined as the optimal policy $\pi^*$ for the induced MDP **M** where only agent $n$ is present.

This naturally leads to our proposed curriculum: Stage 1 trains a single agent in **M** to achieve a greedy policy, which is then used for initialization in **MG** in Stage 2. Next we explain in detail how to leverage the structure of decentralized MARL to bridge the two curriculum stages.

## 4.4 Function augmentation for multi-goal curriculum

In Markov games with decentralized execution, an agent's observation space decomposes into $\mathcal{O}^n = \mathcal{O}_{\text{self}}^n \cup \mathcal{O}_{\text{others}}^n$, where $o_{\text{self}}^n \in \mathcal{O}_{\text{self}}^n$ captures the agent's own properties, which must be observable by the agent for closed-loop control, while $o_{\text{others}}^n \in \mathcal{O}_{\text{others}}^n$ is the agent's egocentric observation of other agents. In our work, egocentric observations are private and not accessible by other agents (Pynadath and Tambe, 2002). Similarly, global state $s$ decomposes into $s := (s_{\text{env}}, s^n, s^{-n})$, where

---

[2]We provide a synthetic example to aid intuition in Appendix D

[3]Environments include: discrete 2D worlds, continuous 3D physics simulators, StarCraft II, transportation tasks, 3D first-person multiplayer games, etc. Exceptions are settings where a single task is purely defined by inter-agent communication, but these are not multi-goal Markov games.

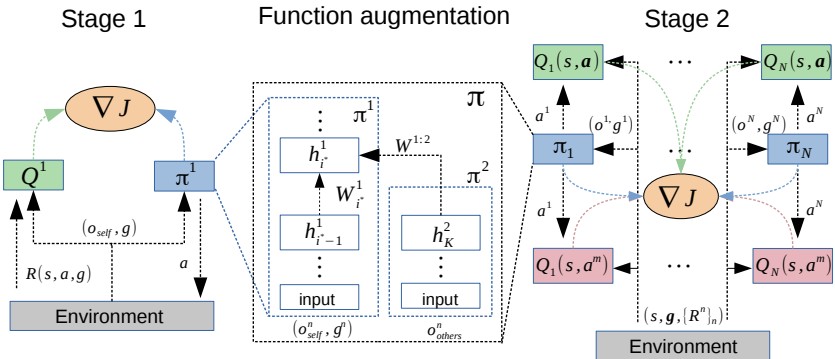

Figure 1: In Stage 1, $Q^1$ and $\pi^1$ learn to achieve multiple goals in a single-agent environment. Between Stage 1 and 2, $\pi$ is constructed from the trained $\pi^1$ and a new module $\pi^2$ according to ( same construction is done for $Q_n(s, \mathbf{a})$ and $Q_n(s, a^m)$, not shown). In the multi-agent environment of Stage 2, these augmented functions are instantiated for each of $N$ agents (with parameter-sharing).

$s_{\text{env}}$ is environment information not specific to any agent (e.g., position of a landmark), and $s^n$ captures agent $n$'s information. While this decomposition is implicitly available in a wide range of complex multi-agent environments (Bansal *et al.*, 2018; Foerster *et al.*, 2018; Lowe *et al.*, 2017; Rashid *et al.*, 2018; Liu *et al.*, 2019; Jaderberg *et al.*, 2019), we explicitly use it to implement our curriculum. In Stage 1, as the ability to process $o_{\text{others}}^n$ and $s^{-n}$ is unnecessary, we reduce the input space of policy and value functions, thereby reducing the number of trainable parameters and lowering the computation cost. In Stage 2, we restore Stage 1 parameters and activate new modules to process additional inputs $o_{\text{others}}^n$ and $s^{-n}$. This augmentation is especially suitable for efficiently learning the credit function (1) and global Q-function, since $Q(s, a)$ can be augmented into both $Q_n^{\boldsymbol{\pi}}(s, \mathbf{a})$ and $Q_n^{\boldsymbol{\pi}}(s, a^m)$, as explained below.

## 4.5 A COMPLETE INSTANTIATION OF CM3

We combine the preceding components to create CM3, using deep neural networks for function approximation (Figure 1 and Algorithm 1). Without loss of generality, we assume parameter-sharing (Foerster *et al.*, 2018) among homogeneous agents with goals as input (Schaul *et al.*, 2015). The inhomogeneous case can be addressed by $N$ actor-critics. Drawing from multi-task learning (Taylor and Stone, 2009), we sample goal(s) in each episode for the agent(s), to train one model for all goals.

**Stage 1.** We train an actor $\pi^1(a|o, g)$ and critic $Q^1(s^1, a, g)$ to convergence according to (4) and (5) in the induced MDP with $N = 1$ and random goal sampling (see Appendix J). This uses orders of magnitude fewer samples than for the full multi-agent environment—compare Figure 6 with Figure 5.

**Stage 2.** The Markov game is instantiated with all $N$ agents. We restore the trained $\pi^1$ parameters, instantiate a second neural network $\pi^2$ for agents to process $o_{\text{others}}^n$, and connect the output of $\pi^2$ to a selected hidden layer of $\pi^1$. Concretely, let $h_i^1 \in \mathbb{R}^{m_i}$ denote hidden layer $i \leq L$ with $m_i$ units in an $L$-layer network $\pi^1$, connected to layer $i-1$ via $h_i^1 = f(W_i^1 h_{i-1}^1)$ with $W_i^1 \in \mathbb{R}^{m_i \times m_{i-1}}$ and nonlinear activation $f$. Stage 2 introduces a $K$-layer network $\pi^2(o_{\text{others}}^n)$ with outputs $h_K^2 \in \mathbb{R}^{m_K}$, chooses a layer[4] $i^*$ of $\pi^1$, and augments $h_{i^*}^1$ to be $h_{i^*}^1 = f(W_{i^*}^1 h_{i^*-1}^1 + W^{1:2} h_K^2)$ with $W^{1:2} \in \mathbb{R}^{m_{i^*} \times m_K}$. Being restored from Stage 1, not re-initialized, hidden layers $i < i^*$ begin with the ability to process $(o_{\text{self}}^n, g^n)$, while the new weights in $\pi^2$ and $W^{1:2}$ specifically learn the effect of surrounding agents. Higher layers $i \geq i^*$ that already take greedy actions to achieve goals in Stage 1 must now do so while cooperating to allow other agents' success. This augmentation scheme is simplest for deep policy and value networks using fully-connected or convolutional layers.

The middle panel of Figure 1 depicts the construction of $\pi$ from $\pi^1$ and $\pi^2$. The global $Q^{\boldsymbol{\pi}}(s, \mathbf{a}, g^n)$ is constructed from $Q^1$ similarly: when the input to $Q^1$ is $(s_{\text{env}}, s^n, a^n, g^n)$, a new module takes input $(s^{-n}, a^{-n})$ and connects to a chosen hidden layer of $Q^1$. Credit function $Q^{\boldsymbol{\pi}}(s, a^m, g^n)$ is

---

[4]Setting $i^*$ to be the last hidden layer worked well in our experiments, without needing to tune.

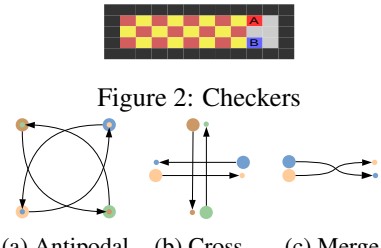

Figure 2: Checkers

(a) Antipodal    (b) Cross    (c) Merge

Figure 3: Cooperative navigation

Figure 4: Agent sedans must perform double lane merge to reach goal lanes. SUMO controls yellow sedans and trucks. Policy generalization was tested on such traffic conditions.

augmented from a copy of $Q^1$, such that when $Q^1$ inputs are $(s_{\text{env}}, s^n, a^m, g^n)$, the new module's inputs are $(s^m, s^{-n})$.[5] We train the policy using (5), train the credit function with loss (4), and train the global Q-function with the joint-action analogue of (4).

# 5   EXPERIMENTAL SETUP

We investigated the performance and robustness of CM3 versus existing methods on diverse and challenging multi-goal MARL environments: cooperative navigation in difficult formations, double lane merge in autonomous driving, and strategic cooperation in a Checkers game. We evaluated ablations of CM3 on all domains. We describe key setup here, with full details in Appendices G to J[6].

**Cooperative navigation:** We created three variants of the cooperative navigation scenario in Lowe *et al.* (2017), where $N$ agents cooperate to reach a set of targets. We increased the difficulty by giving each agent only an individual reward based on distance to its designated target, not a global team reward, but initial and target positions require complex cooperative maneuvers to avoid collision penalties (Figure 3). Agents observe relative positions and velocities (details in Appendix G.1). **SUMO:** Previous work modeled autonomous driving tasks as MDPs in which all other vehicles do not learn to respond to a single learning agent (Isele *et al.*, 2018; Kuefler *et al.*, 2017). However, real-world driving requires cooperation among different drivers' with personal goals. Built in the SUMO traffic simulator with sublane resolution (Lopez *et al.*, 2018), this experiment requires agent vehicles to learn double-merge maneuvers to reach goal lane assignments (Figure 4). Agents have limited field of view and receive sparse rewards (Appendix G.2). **Checkers:** We implemented a challenging strategic game (Appendix G.3, an extension of Sunehag *et al.* (2018)), to investigate whether CM3 is beneficial even when an agent cannot maximize its reward in the absence of another agent. In a gridworld with red and yellow squares that disappear when collected (Figure 2), Agent A receives +1 for red and -0.5 for yellow; Agent B receives -0.5 for red and +1 for yellow. Both have a limited 5x5 field of view. The global optimum requires each agent to clear the path for the other.

**Algorithm implementations.** We describe key points here, leaving complete architecture details and hyperparameter tables to Appendices H and I. **CM3:** Stage 1 is defined for each environment as follows (Appendix G): in cooperative navigation, a single particle learns to reach any specified landmark; in SUMO, a car learns to reach any specified goal lane; in Checkers, we alternate between training one agent as A and B. Appendix H describes function augmentation in Stage 2 of CM3. **COMA** (Foerster *et al.*, 2018): the joint goal $\mathbf{g}$ and total reward $\sum_n R^n$ can be used to train COMA's global Q function, which receives input $(s, o^n, g^n, n, a^{-n}, g^{-n})$. Each output node $i$ represents $Q(s, a^n = i, a^{-n}, \mathbf{g})$. **IAC** (Tan, 1993; Foerster *et al.*, 2018): IAC trains each agent's actor and critic independently, using the agent's own observation. The TD error of value function $V(o^n, g^n)$ is used in a standard policy gradient (Sutton *et al.*, 2000). **QMIX** (Rashid *et al.*, 2018): we used the original hypernetwork, giving all goals to the mixer and individual goals to each agent network. We used a manual coordinate descent on exploration and learning rate hyperparameters, including values reported in the original works. We ensured the number of trainable parameters are similar among all methods, up to method-specific architecture requirements for COMA and QMIX.

---

[5]Input $s^m$ is needed for disambiguation, so that input action $a^m$ is associated with agent $m$.
[6]Code for all experiments is available at https://github.com/011235813/cm3.

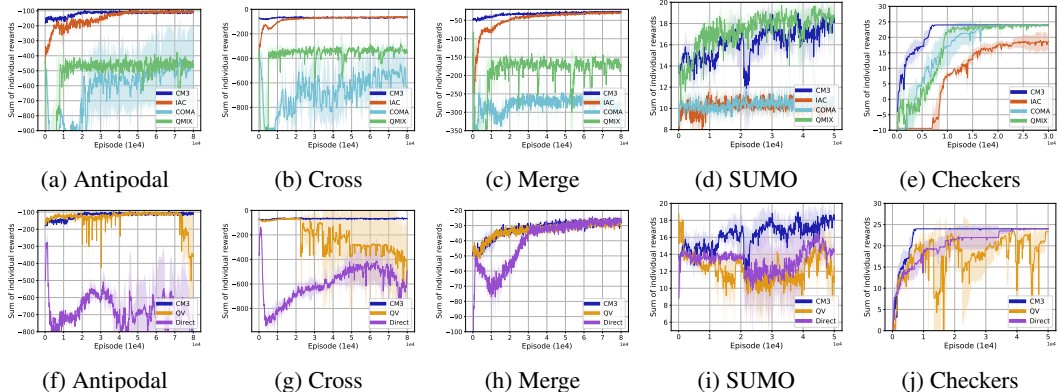

Figure 5: a-e: Comparison against baselines in cooperative navigation (a-c), SUMO (d), Checkers (e). f-j: Comparison against ablations. Average and standard deviation (shaded) of 10 evaluation episodes conducted every 100 training episodes, across 3 independent runs.

**Ablations.** We conducted ablation experiments in all domains. To discover the speedup from the curriculum with function augmentation, we trained the full Stage 2 architecture of CM3 (labeled as **Direct**) without first training components $\pi^1$ and $Q^1$ in an induced MDP. To investigate the benefit of the new credit function and multi-goal policy gradient, we trained an ablation (labeled **QV**) with advantage function $A_n^\pi(s, \mathbf{a}) := Q_n^\pi(s, \mathbf{a}) - V_n^\pi(s)$, where credit assignment between action-goal pairs is lost. QV uses the same $\pi^1$, $Q^1$, and function augmentation as CM3.

## 6 RESULTS AND DISCUSSIONS

CM3 finds optimal or near-optimal policies significantly faster than IAC and COMA on all domains, and performs significantly higher than QMIX in four out of five. We report absolute runtime in Appendix F and account for CM3's Stage 1 episodes (Appendix J) when comparing sample efficiency.

**Main comparison.** Over all **cooperative navigation** scenarios (Figures 5a to 5c), CM3 (with 1k episodes in Stage 1) converged more than 15k episodes faster than IAC. IAC reached the same final performance as CM3 because dense individual rewards simplifies the learning problem for IAC's fully decentralized approach, but CM3 benefited significantly from curriculum learning, as evidenced by comparison to "Direct" in Figure 5f. QMIX and COMA settled at suboptimal behavior. Both learn global critics that use all goals as input, in contrast to CM3 and IAC that process each goal separately. This indicates the difficulty of training agents for individual goals under a purely global approach. While COMA was shown to outperform IAC in SC2 micromanagement where IAC must learn from a single team reward (Foerster *et al.*, 2018), our IAC agents have access to individual rewards that resolve the credit assignment issue and improve performance (Singh *et al.*, 2019). In **SUMO** (Figure 5d), CM3 and QMIX found cooperative solutions with performances within the margin of error, while COMA and IAC could not break out of local optima where vehicles move straight but do not perform merge maneuvers. Since initial states force agents into the region of state space requiring cooperation, credit assignment rather than exploration is the dominant challenge, which CM3 addressed via the credit function, as evidenced in Figure 5i. IAC underperformed because SUMO requires a longer sequence of cooperative actions and gave much sparser rewards than the "Merge" scenario in cooperative navigation. We also show that centralized training of merely two decentralized agents allows them to generalize to settings with much heavier traffic (Appendix E). In **Checkers** (Figure 5e), CM3 (with 5k episodes in Stage 1) converged 10k episodes faster than COMA and QMIX to the global optimum with score 24. Both exploration of the combinatorially large joint trajectory space and credit assignment for path clearing are challenges that CM3 successfully addressed. COMA only solved Checkers among all domains, possibly because the small bounded environment alleviates COMA's difficulty with individual goals in large state spaces. IAC underperformed all centralized learning methods because cooperative actions that give no instantaneous reward are hard for selfish agents to discover in Checkers. These results demonstrate CM3's ability to attain individual goals and find cooperative solutions in diverse multi-agent systems.

**Ablations.** The significantly better performance of CM3 versus "Direct" (Figures 5f to 5j) shows that learning individual goal attainment prior to learning multi-agent cooperation, and initializing Stage 2 with Stage 1 parameters, are crucial for improving learning speed and stability. It gives evidence that while global action-value and credit functions may be difficult to train from scratch, function augmentation significantly eases the learning problem. While "QV" initially learns quickly to attain individual goals, it does so at the cost of frequent collisions, higher variance, and inability to maintain a cooperative solution, giving clear evidence for the necessity of the credit function.

## 7 CONCLUSION

We presented CM3, a general framework for cooperative multi-goal MARL. CM3 addresses the need for efficient exploration to learn both individual goal attainment and cooperation, via a two-stage curriculum bridged by function augmentation. It achieves local credit assignment between action and goals using a credit function in a multi-goal policy gradient. In diverse experimental domains, CM3 attains significantly higher performance, faster learning, and overall robustness than existing MARL methods, displaying strengths of both independent learning and centralized credit assignment while avoiding shortcomings of existing methods. Ablations demonstrate each component is crucial to the whole framework. Our results motivate future work on analyzing CM3's theoretical properties and generalizing to inhomogeneous systems or settings without known goal assignments.

ACKNOWLEDGMENTS

JY thanks Rakshit Trivedi (Georgia Institute of Technology), Ahmad Beirami (Facebook AI), and Peter Sunehag (DeepMind) for detailed and helpful feedback on this work.

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

## A  ALGORITHM

---

**Algorithm 1** Cooperative multi-goal multi-stage multi-agent reinforcement learning (CM3)

---

1: **for** curriculum stage $c = 1$ to 2 **do**
2:     **if** $c = 1$ **then**
3:         Set number of agents $N = 1$
4:         Initialize Stage 1 main networks $Q_g := Q = Q^1, \pi := \pi^1$ with parameters $\theta_{Q^1}, \theta_{\pi^1}$
5:         Initialize target networks with $\theta'_{\pi^1}, \theta'_{Q^1}$
6:     **else if** $c = 2$ **then**
7:         Instantiate $N > 1$ agents
8:         Construct global $Q_g := Q_n^{\boldsymbol{\pi}}(s, \mathbf{a}) = \{Q^1, Q_g^2\}$, credit function $Q_c := Q_n^{\boldsymbol{\pi}}(s, a^m) = \{Q^1, Q_c^2\}$ and $\pi := \{\pi^1, \pi^2\}$ using function augmentation with parameters $\theta_{Q_g}, \theta_{Q_c}, \theta_\pi$
9:         Initialize target networks with $\theta'_{Q_g}, \theta'_{Q_c}, \theta'_\pi$
10:         Restore values of trained parameters $\theta_{Q^1}, \theta_{\pi^1}$ into the respective subsets of $\theta_{Q_g}, \theta_{Q_c}, \theta_\pi$
11:     **end if**
12:     Set all target network weights to equal main networks weights
13:     Initialize exploration parameter $\epsilon = \epsilon_{\text{start}}$ and empty replay buffer $B$
14:     **for** each training episode $e = 1$ to $E$ **do**
15:         Assign goal(s) $g_e^n$ to agent(s) according to given distribution
16:         Get initial state $s_1$ and observation(s) $\mathbf{o}_1$
17:         **for** $t = 1$ to $T$ **do** *// execute policies in environment*
18:             Sample action $a_t^n \sim \pi(a_t^n | o_t^n; \theta_\pi, \epsilon)$ for each agent.
19:             Execute action(s) $\mathbf{a}_t$, receive $\{r_t^n\}_n, s_{t+1}$, and $\mathbf{o}_{t+1}$
20:             Store $(s_t, \mathbf{o}_t, \mathbf{g}_e, \mathbf{a}_t, \{r_t^n\}_n, R_t^g, s_{t+1}, \mathbf{o}_{t+1})$ into $B$
21:             $s_t \leftarrow s_{t+1}, \mathbf{o}_t \leftarrow \mathbf{o}_{t+1}$
22:         **end for**
23:         **if** e mod $E_{\text{train}} = 0$ **then**
24:             **for** epochs $1 \ldots K$ **do** *// conduct training*
25:                 Sample minibatch of $S$ transitions $(s_i, \mathbf{o}_i, \mathbf{g}_i, \mathbf{a}_i, \{r_i^n\}_n, s_{i+1}, \mathbf{o}_{i+1})$ from $B$
26:                 Compute global target for all $n$: $x_i^n = r_i^n + \gamma Q(s_{i+1}, \mathbf{a}_{i+1}, g_i^n; \theta'_{Q_g})|_{\mathbf{a}_{i+1} \sim \boldsymbol{\pi}'}$
27:                 Gradient descent on $\mathcal{L}(\theta_{Q_g}) = \frac{1}{S} \sum_i \frac{1}{N} \sum_{n=1}^N \left(x_i^n - Q(s_i, \mathbf{a}_i, g_i^n; \theta_{Q_g})\right)^2$
28:                 **if** c = 1 **then**
29:                     $A^\pi(s_i, a_i) = Q^1(s_i, a_i, g_i; \theta_{Q^1}) - \sum_{\hat{a}_i} \pi(\hat{a}_i | o_i, g_i) Q^1(s_i, \hat{a}_i, g_i; \theta_{Q^1})$
30:                 **else if** c = 2 **then**
31:                     $\forall m, n \in [1..N]$, compute target $y_i^n = r_i^n + \gamma Q(s_{i+1}, a_{i+1}^m, g^n; \theta'_{Q_c})|_{a_{i+1}^m \sim \pi'^m}$
32:                     Minimize (4): $\mathcal{L}(\theta_{Q_c}) = \frac{1}{S} \sum_i \frac{1}{N^2} \sum_{n=1}^N \sum_{m=1}^N (y_i^n - Q(s_i, a_i^m, g_i^n; \theta_{Q_c}))^2$
33:                     $A_{n,m}^{\boldsymbol{\pi}}(s_i, \mathbf{a}_i) := Q(s_i, \mathbf{a}_i, g_i^n; \theta_{Q_g}) - \sum_{\hat{a}^m} \pi(\hat{a}^m) Q(s_i, \hat{a}^m, g_i^n; \theta_{Q_c})$
34:                 **end if**
35:                 $\nabla_{\theta_\pi} J(\boldsymbol{\pi}) = \frac{1}{S} \sum_i \sum_{m,n=1}^N (\nabla_{\theta_\pi} \log \pi(a_i^m | o_i^m, g_i^m)) A_{n,m}^{\boldsymbol{\pi}}(s_i, \mathbf{a}_i)$
36:                 Update policy: $\theta_\pi \leftarrow \theta_\pi + \beta \nabla_{\theta_\pi} J(\boldsymbol{\pi})$
37:                 Update all target network parameters using: $\theta' \leftarrow \tau\theta + (1-\tau)\theta'$
38:                 Reset buffer $B$
39:             **end for**
40:         **end if**
41:         If $\epsilon > \epsilon_{\text{end}}$, then $\epsilon \leftarrow \epsilon - \epsilon_{\text{step}}$
42:     **end for**
43: **end for**

---

Off-policy training with a large replay buffer allows RL algorithms to benefit from less correlated transitions (Silver *et al.*, 2014; Lillicrap *et al.*, 2016). The algorithmic modification for off-policy training is to maintain a circular replay buffer that does not reset (i.e. remove line 38), and conduct training (lines 24-41) while executing policies in the environment (lines 17-22). Despite introducing bias in MARL, we found that off-policy training benefited CM3 in SUMO and Checkers.

# B  DERIVATIONS

## B.1  PROPOSITION 1

By stationarity and relabeling $t$, the credit function can be written:

$$Q_n^{\boldsymbol{\pi}}(s, a^m) := \mathbb{E}_{\boldsymbol{\pi}}\Big[\sum_{t=0}^{\infty}\gamma^t R(s_t, \mathbf{a}_t, g^n) \,\Big|\, s_0 = s, a_0^m = a^m\Big] = \mathbb{E}_{\boldsymbol{\pi}}\Big[\sum_{t=1}^{\infty}\gamma^{t-1} R(s_t, \mathbf{a}_t, g^n) \,\Big|\, s_1 = s, a_1^m = a^m\Big]$$

Using the law of iterated expectation, the credit function satisfies the Bellman expectation equation (2):

$$Q_n^{\boldsymbol{\pi}}(s, a^m) = \mathbb{E}_{\boldsymbol{\pi}}\Big[\sum_{t=0}^{\infty}\gamma^t R(s_t, \mathbf{a}_t, g^n) \,\Big|\, s_0 = s, a_0^m = a^m\Big]$$

$$= \mathbb{E}_{\boldsymbol{\pi}}\Big[R(s_0, \mathbf{a}_0, g^n) + \sum_{t=1}^{\infty}\gamma^t R(s_t, \mathbf{a}_t, g^n) \,\Big|\, s_0 = s, a_0^m = a^m\Big]$$

$$= \mathbb{E}_{s_1, a_1^m | s_0, a_0, \boldsymbol{\pi}}\Big[\mathbb{E}_{\boldsymbol{\pi}}\Big[R(s_0, \mathbf{a}_0, g^n) + \sum_{t=1}^{\infty}\gamma^t R(s_t, \mathbf{a}_t, g^n) \,\Big|\, s_0 = s, a_0^m = a^m, s_1 = s', a_1^m = \hat{a}^m\Big] \,\Big|\, s_0 = s, a_0^m = a^m\Big]$$

$$= \mathbb{E}_{s_1, a_1^m | s_0, a_0, \boldsymbol{\pi}}\Big[\sum_{a^{-m}}\boldsymbol{\pi}(a^{-m}|s, \mathbf{g}^{-m})R(s, (a^m, a^{-m}), g^n)$$

$$+ \mathbb{E}_{\boldsymbol{\pi}}\Big[\sum_{t=1}^{\infty}\gamma^t R(s_t, \mathbf{a}_t, g^n) \,\Big|\, s_0 = s, a_0^m = a^m, s_1 = s', a_1^m = \hat{a}^m\Big] \,\Big|\, s_0 = s, a_0^m = a^m\Big]$$

$$= \sum_{a^{-m}}\boldsymbol{\pi}(a^{-m}|s, \mathbf{g}^{-m})R(s, (a^m, a^{-m}), g^n)$$

$$+ \mathbb{E}_{s_1, a_1^m | s_0, a_0, \boldsymbol{\pi}}\Big[\mathbb{E}_{\boldsymbol{\pi}}\Big[\sum_{t=1}^{\infty}\gamma^t R(s_t, \mathbf{a}_t, g^n) \,\Big|\, s_0 = s, a_0^m = a^m, s_1 = s', a_1^m = \hat{a}^m\Big] \,\Big|\, s_0 = s, a_0^m = a^m\Big]$$

$$= \sum_{a^{-m}}\boldsymbol{\pi}(a^{-m}|s, \mathbf{g}^{-m})R(s, (a^m, a^{-m}), g^n)$$

$$+ \sum_{a^{-m}}\boldsymbol{\pi}(a^{-m}|s, \mathbf{g}^{-m})\sum_{s'}P(s'|s, (a^m, a^{-m}))\sum_{\hat{a}^m}\pi(\hat{a}^m|o^m(s'))\mathbb{E}_{\boldsymbol{\pi}}\Big[\sum_{t=1}^{\infty}\gamma^t R(s_t, \mathbf{a}_t, g^n) \,\Big|\, s_1 = s', a_1^m = \hat{a}^m\Big]$$

$$= \sum_{a^{-m}}\boldsymbol{\pi}(a^{-m}|s, \mathbf{g}^{-m})\Big[R(s, (a^m, a^{-m}), g^n)$$

$$+ \gamma\sum_{s'}P(s'|s, (a^m, a^{-m}))\sum_{\hat{a}^m}\pi(\hat{a}^m|o^m(s'))\mathbb{E}_{\pi}\Big[\sum_{t=1}^{\infty}\gamma^{t-1} R(s_t, \mathbf{a}_t, g^n) \,\Big|\, s_1 = s', a_1^m = \hat{a}^m\Big]\Big]$$

$$= \sum_{a^{-m}}\boldsymbol{\pi}(a^{-m}|s, \mathbf{g}^{-m})\Big[R(s, (a^m, a^{-m}), g^n) + \gamma\sum_{s'}P(s'|s, (a^m, a^{-m}))\sum_{\hat{a}^m}\pi^m(\hat{a}^m|o^m(s'))Q_n^{\boldsymbol{\pi}}(s', \hat{a}^m)\Big]$$

$$= \mathbb{E}_{\boldsymbol{\pi}}\Big[R(s_t, \mathbf{a}_t, \mathbf{g}^n) + \gamma Q_n^{\boldsymbol{\pi}}(s_{t+1}, a_{t+1}^m)\Big|s_t = s, a_t^m = a^m\Big]$$

$\square$

The goal-specific joint value function is the marginal of the credit function:

$$V_n^{\boldsymbol{\pi}}(s) = \mathbb{E}_{\boldsymbol{\pi}}\Big[\sum_{t=0}^{\infty}\gamma^t R(s_t, \mathbf{a}_t, g^n) \,\Big|\, s_0 = s\Big]$$

$$= \mathbb{E}_{a_0^m | s_0, \boldsymbol{\pi}}\Big[\mathbb{E}_{\boldsymbol{\pi}}\Big[\sum_{t=0}^{\infty}\gamma^t R(s_t, \mathbf{a}_t, g^n) \,\Big|\, s_0 = s, a_0^m = a^m\Big] \,\Big|\, s_0 = s\Big]$$

$$= \sum_{a^m}\pi(a^m|o^m(s), g^m)Q_n^{\boldsymbol{\pi}}(s, a^m) \qquad \square$$

The credit function can be expressed in terms of the goal-specific action-value function:

$$V_n^{\boldsymbol{\pi}}(s) = \sum_{a^m} \pi(a^m|o^m, g^m) Q_n^{\boldsymbol{\pi}}(s, a^m) \qquad \text{by (3)}$$

$$V_n^{\boldsymbol{\pi}}(s) = \sum_{\mathbf{a}} \boldsymbol{\pi}(\mathbf{a}|s, \mathbf{g}) Q_n^{\boldsymbol{\pi}}(s, \mathbf{a}) \qquad \text{by (8)}$$

$$= \sum_{a^m} \sum_{a^{-m}} \pi(a^m|o^m, g^m) \boldsymbol{\pi}(a^{-m}|s, g^{-m}) Q_n^{\boldsymbol{\pi}}(s, (a^m, a^{-m}))$$

$$\Rightarrow Q_n^{\boldsymbol{\pi}}(s, a^m) = \sum_{a^{-m}} \boldsymbol{\pi}(a^{-m}|s, g^{-m}) Q_n^{\boldsymbol{\pi}}(s, \mathbf{a}) \qquad \square$$

## B.2 PROPOSITION 2

First we state some elementary relations between global functions $V_n^{\pi}(s)$ and $Q_n^{\pi}(s, \mathbf{a})$. These carry over directly from the case of an MDP, by treating the joint policy $\boldsymbol{\pi}$ as as an effective "single-agent" policy and restricting attention to a single goal $g^n$ (standard derivations are included at the end of this section).

$$Q_n^{\boldsymbol{\pi}}(s, \mathbf{a}) = R(s, \mathbf{a}, g^n) + \gamma \sum_{s'} P(s'|s, \mathbf{a}) V_n^{\boldsymbol{\pi}}(s') \qquad (7)$$

$$V_n^{\boldsymbol{\pi}}(s) = \sum_{\mathbf{a}} \boldsymbol{\pi}(\mathbf{a}|s, \mathbf{g}) Q_n^{\boldsymbol{\pi}}(s, \mathbf{a}) \qquad (8)$$

We follow the proof of the policy gradient theorem (Sutton *et al.*, 2000):

$$\nabla_\theta V_n^{\boldsymbol{\pi}}(s) = \nabla_\theta \sum_{\mathbf{a}} \boldsymbol{\pi}(\mathbf{a}|s, \mathbf{g}) Q_n^{\boldsymbol{\pi}}(s, \mathbf{a})$$

$$= \sum_{\mathbf{a}} \left[ (\nabla_\theta \boldsymbol{\pi}(\mathbf{a}|s, \mathbf{g})) Q_n^{\boldsymbol{\pi}}(s, \mathbf{a}) + \boldsymbol{\pi}(\mathbf{a}|s, \mathbf{g}) \nabla_\theta Q_n^{\boldsymbol{\pi}}(s, \mathbf{a}) \right]$$

$$= \sum_{\mathbf{a}} \left[ (\nabla_\theta \boldsymbol{\pi}(\mathbf{a}|s, \mathbf{g})) Q_n^{\boldsymbol{\pi}}(s, \mathbf{a}) + \boldsymbol{\pi}(\mathbf{a}|s, \mathbf{g}) \nabla_\theta \left( R(s, \mathbf{a}, g^n) + \gamma \sum_{s'} P(s'|s, \mathbf{a}) V_n^{\boldsymbol{\pi}}(s') \right) \right]$$

$$= \sum_{\mathbf{a}} \left[ (\nabla_\theta \boldsymbol{\pi}(\mathbf{a}|s, \mathbf{g})) Q_n^{\boldsymbol{\pi}}(s, \mathbf{a}) + \boldsymbol{\pi}(\mathbf{a}|s, \mathbf{g}) \gamma \sum_{s'} P(s'|s, \mathbf{a}) \nabla_\theta V_n^{\boldsymbol{\pi}}(s') \right]$$

$$= \sum_{\hat{s}} \sum_{k=0}^{\infty} \gamma^k P(s \to \hat{s}, k, \boldsymbol{\pi}) \sum_{\mathbf{a}} (\nabla_\theta \boldsymbol{\pi}(\mathbf{a}|\hat{s}, \mathbf{g})) Q_n^{\boldsymbol{\pi}}(\hat{s}, \mathbf{a}) \quad \text{(by recursively unrolling)}$$

$$\nabla_\theta J_n(\boldsymbol{\pi}) := \nabla_\theta V_n^{\boldsymbol{\pi}}(s_0) = \sum_{s} \sum_{k=0}^{\infty} \gamma^k P(s_0 \to s, k, \boldsymbol{\pi}) \sum_{\mathbf{a}} (\nabla_\theta \boldsymbol{\pi}(\mathbf{a}|s, \mathbf{g})) Q_n^{\boldsymbol{\pi}}(s, \mathbf{a})$$

$$= \sum_{s} \rho^{\boldsymbol{\pi}}(s) \sum_{\mathbf{a}} \boldsymbol{\pi}(\mathbf{a}|s, \mathbf{g}) (\nabla_\theta \log \boldsymbol{\pi}(\mathbf{a}|s, \mathbf{g})) Q_n^{\boldsymbol{\pi}}(s, \mathbf{a})$$

$$= \mathbb{E}_{\boldsymbol{\pi}} \left[ (\nabla_\theta \log \boldsymbol{\pi}(\mathbf{a}|s, \mathbf{g})) Q_n^{\boldsymbol{\pi}}(s, \mathbf{a}) \right] \qquad (9)$$

We can replace $Q_n^{\boldsymbol{\pi}}(s, \mathbf{a})$ by the advantage function $A_n^{\boldsymbol{\pi}}(s, \mathbf{a}) := Q_n^{\boldsymbol{\pi}}(s, \mathbf{a}) - V_n^{\boldsymbol{\pi}}(s)$, which does not change the expectation in Equation (9) because:

$$\mathbb{E}_{\boldsymbol{\pi}} \left[ \nabla_\theta \log \boldsymbol{\pi}(\mathbf{a}|s, \mathbf{g}) V_n^{\boldsymbol{\pi}}(s) \right] = \sum_{s} \rho^{\boldsymbol{\pi}}(s) \sum_{\mathbf{a}} \boldsymbol{\pi}(\mathbf{a}|s, \mathbf{g}) \nabla_\theta \log \boldsymbol{\pi}(\mathbf{a}|s, \mathbf{g}) V_n^{\boldsymbol{\pi}}(s)$$

$$= \sum_{s} \rho^{\boldsymbol{\pi}}(s) V_n^{\boldsymbol{\pi}}(s) \nabla_\theta \sum_{\mathbf{a}} \boldsymbol{\pi}(\mathbf{a}|s, \mathbf{g}) = 0$$

So the gradient (9) can be written

$$\nabla_\theta J_n(\boldsymbol{\pi}) = \mathbb{E}_{\boldsymbol{\pi}} \left[ \left( \nabla_\theta \sum_{m=1}^{N} \log \pi(a^m|o^m, g^m) \right) \left( Q_n^{\boldsymbol{\pi}}(s, \mathbf{a}) - V_n^{\boldsymbol{\pi}}(s) \right) \right] \qquad (10)$$

Recall that from (3), for any choice of agent label $k \in [1..N]$:

$$V_n^{\boldsymbol{\pi}}(s) = \sum_{a^k} \pi(a^k|o^k, g^k) Q_n^{\boldsymbol{\pi}}(s, a^k) \tag{11}$$

Then substituting (3) into (10):

$$\nabla_\theta J_n(\boldsymbol{\pi}) = \mathbb{E}_{\boldsymbol{\pi}}\left[\left(\nabla_\theta \sum_{m=1}^N \log \pi(a^m|o^m, g^m)\right) A_{n,k}^{\boldsymbol{\pi}}(s, \mathbf{a})\right] \tag{12}$$

$$A_{n,k}^{\boldsymbol{\pi}}(s, \mathbf{a}) := Q_n^{\boldsymbol{\pi}}(s, \mathbf{a}) - \sum_{\hat{a}^k} \pi(\hat{a}^k|o^k, g^k) Q_n^{\boldsymbol{\pi}}(s, \hat{a}^k) \tag{13}$$

Now notice that the choice of $k$ in (13) is completely arbitrary, since (3) holds for any $k \in [1..N]$. Therefore, it is valid to distribute $A_{n,k}^{\boldsymbol{\pi}}(s, \mathbf{a})$ into the summation in (12) *using the summation index $m$ instead of $k$*. Further summing (12) over all $n$, we arrive at the result of Proposition 2:

$$\nabla_\theta J(\boldsymbol{\pi}) = \mathbb{E}_{\boldsymbol{\pi}}\left[\sum_{m=1}^N \sum_{n=1}^N \left(\nabla_\theta \log \pi(a^m|o^m, g^m)\right) A_{n,m}^{\boldsymbol{\pi}}(s, \mathbf{a})\right]$$

$$A_{n,m}^{\boldsymbol{\pi}}(s, \mathbf{a}) := Q_n^{\boldsymbol{\pi}}(s, \mathbf{a}) - \sum_{\hat{a}^m} \pi(\hat{a}^m|o^m, g^m) Q_n^{\boldsymbol{\pi}}(s, \hat{a}^m) \qquad \square$$

The relation between $V_n^\pi(s)$ and $Q_n^\pi(s, \mathbf{a})$ in (7) and (8) are derived as follows:

$$Q_n^{\boldsymbol{\pi}}(s, \mathbf{a}) := \mathbb{E}_\pi\left[\sum_t \gamma^t R(s_t, \mathbf{a}_t, g^n) \mid s_0 = s, \mathbf{a}_0 = \mathbf{a}\right]$$

$$= \mathbb{E}_\pi\left[R(s_0, \mathbf{a}_0, g^n) + \sum_{t=1}^\infty \gamma^t R(s_t, \mathbf{a}_t, g^n) \mid s_0 = s, \mathbf{a}_0 = \mathbf{a}\right]$$

$$= R(s, \mathbf{a}, g^n) + \mathbb{E}_{s_1|s_0, \mathbf{a}_0, \boldsymbol{\pi}}\left[\mathbb{E}_\pi\left[\sum_{t=1}^\infty R(s_t, \mathbf{a}_t, g^n) \mid s_0 = s, \mathbf{a}_0 = \mathbf{a}, s_1 = s'\right] \mid s_0 = s, \mathbf{a}_0 = \mathbf{a}\right]$$

$$= R(s, \mathbf{a}, g^n) + \gamma \sum_{s'} P(s'|s, \mathbf{a}) \mathbb{E}_{\boldsymbol{\pi}}\left[\sum_{t=1}^\infty \gamma^{t-1} R(s_t, \mathbf{a}_t, g^n) \mid s_1 = s'\right]$$

$$= R(s, \mathbf{a}, g^n) + \gamma \sum_{s'} P(s'|s, \mathbf{a}) V_n^{\boldsymbol{\pi}}(s')$$

$$V_n^{\boldsymbol{\pi}}(s) := \mathbb{E}_{\boldsymbol{\pi}}\left[\sum_{t=0}^\infty \gamma^t R(s_t, \mathbf{a}_t, g^n) \mid s_0 = s\right]$$

$$= \mathbb{E}_{\mathbf{a}_0|s_0, \boldsymbol{\pi}}\left[\mathbb{E}_{\boldsymbol{\pi}}\left[\sum_{t=0}^\infty \gamma^t R(s_t, \mathbf{a}_t, g^n) \mid s_0 = s, \mathbf{a}_0 = \mathbf{a}\right] \mid s_0 = s\right]$$

$$= \sum_{\mathbf{a}} \boldsymbol{\pi}(\mathbf{a}|s, \mathbf{g}) \mathbb{E}_{\boldsymbol{\pi}}\left[\sum_{t=0}^\infty \gamma^t R(s_t, \mathbf{a}_t, g^n) \mid s_0 = s, \mathbf{a}_0 = \mathbf{a}\right]$$

$$= \sum_{\mathbf{a}} \boldsymbol{\pi}(\mathbf{a}|s, \mathbf{g}) Q_n^{\boldsymbol{\pi}}(s, \mathbf{a}) \qquad \square$$

## C    VARIANCE

### C.1    VARIANCE OF COMA GRADIENT.

Let $Q := Q^{\boldsymbol{\pi}}(s, \mathbf{a}, \mathbf{g})$ denote the centralized Q function, let $\pi(a^n) := \pi(a^n | o^n, g^n)$ denote a single agent's policy, and let $\pi(a^{-n}) := \pi(a^{-n} | o^{-n}, g^{-n})$ denote the other agents' joint policy.

In cooperative multi-goal MARL, the direct application of COMA has the following gradient.

$$\nabla_\theta J = \mathbb{E}\Big[\sum_n \nabla_\theta \log \pi(a^n | o^n, g^n)\big(Q - b_n(s, a^{-n}, \mathbf{g})\big)\Big]$$

$$b_n(s, a^{-n}, \mathbf{g}) := \sum_{\hat{a}^n} \pi(\hat{a}^n | o^n, g^n) Q^{\boldsymbol{\pi}}(s, \hat{a}^n, a^{-n}, \mathbf{g})$$

Define the following:

$$z_n := \nabla_\theta \log \pi(a^n | o^n, g^n)$$

$$f_n := \nabla_\theta \log \pi(a^n | o^n, g^n)\big(Q - b_n(s, a^{-n})\big) = z_n\big(Q - b_n(s, a^{-n}, \mathbf{g})\big)$$

Define $M_{nm} := \mathbb{E}_{\boldsymbol{\pi}}[f_n]^T \mathbb{E}_{\boldsymbol{\pi}}[f_m]$ and let $M := \sum_{n,m} M_{nm}$. Then we have $M_{nm} = \mathbb{E}_{\boldsymbol{\pi}}[z_n Q]^T \mathbb{E}_{\boldsymbol{\pi}}[z_m Q]$ since

$$\mathbb{E}_{\boldsymbol{\pi}}[z_n b_n] = \mathbb{E}_{\boldsymbol{\pi}}\Big[\sum_s \rho^{\boldsymbol{\pi}}(s) \sum_{\mathbf{a}} \boldsymbol{\pi}(\mathbf{a}|s, \mathbf{g}) \nabla_\theta \log \pi(a^n | o^n, g^n) b_n(s, a^{-n}, \mathbf{g})\Big]$$

$$= \sum_s \rho^{\boldsymbol{\pi}}(s) \sum_{a^{-n}} \pi^{-n}(a^{-n} | o^{-n}, g^{-n}) \sum_{a^n} \pi(a^n | o^n, g^n) \nabla_\theta \log \pi(a^n | o^n, g^n) b_n(s, a^{-n}, \mathbf{g})$$

$$= \sum_s \rho^{\boldsymbol{\pi}}(s) \sum_{a^{-n}} \pi^{-n}(a^{-n} | o^{-n}, g^{-n}) \sum_{a^n} \nabla_\theta \pi(a^n | o^n, g^n) b_n(s, a^{-n}, \mathbf{g})$$

$$= \sum_s \rho^{\boldsymbol{\pi}}(s) \sum_{a^{-n}} \pi^{-n}(a^{-n} | o^{-n}, g^{-n}) b_n(s, a^{-n}, \mathbf{g}) \nabla_\theta \sum_{a^n} \pi(a^n | o^n, g^n) = 0$$

Since the COMA gradient is $\mathbb{E}_{\boldsymbol{\pi}}[\sum_{n=1}^N f_n]$. its variance can be derived to be (Wu *et al.*, 2018):

$$\mathrm{Var}(\sum_{n=1}^N f_n) = \sum_n \mathbb{E}_{\boldsymbol{\pi}}\Big[z_n^T z_n Q^2 - 2b_n z_n^T z_n Q + b_n^2 z_n^T z_n\Big]$$

$$+ \sum_n \sum_{m \neq n} \mathbb{E}_{\boldsymbol{\pi}}\Big[z_n^T z_m (Q - b_n)(Q - b_m)\Big] - M$$

### C.2    VARIANCE OF THE CM3 GRADIENT

For convenience, let $Q_n := Q_n^{\boldsymbol{\pi}}(s, \mathbf{a}) = Q^{\boldsymbol{\pi}}(s, \mathbf{a}, g^n)$ denote the global Q function for goal $g^n$, and let $\pi(a^m) := \pi(a^m | o^m, g^m)$. The CM3 gradient can be rewritten as

$$\nabla_\theta J(\boldsymbol{\pi}) = \mathbb{E}_{\boldsymbol{\pi}}\Big[\sum_{n=1}^N \sum_{m=1}^N \nabla_\theta \log \pi(a^m)\big(Q_n - b_{nm}(s)\big)\Big]$$

$$b_{nm}(s) := \sum_{\hat{a}^m} \pi(\hat{a}^m) Q_n^{\boldsymbol{\pi}}(s, \hat{a}^m)$$

As before, $z_m := \nabla_\theta \log \pi(a^m)$. Define $h_{nm} := z_m(Q_n - b_{nm}(s))$ and let $h_n := \sum_m h_{nm}$. Then the variance is

$$\mathrm{Var}(\sum_n h_n) = \sum_n \mathrm{Var}(h_n) + \sum_n \sum_{m \neq n} \mathrm{Cov}(h_n, h_m)$$

$$= \sum_n \Big(\sum_m \mathrm{Var}(h_{nm}) + \sum_m \sum_{k \neq m} \mathrm{Cov}(h_{nm}, h_{nk})\Big) + \sum_n \sum_{m \neq n} \mathrm{Cov}(h_n, h_m)$$

## D    EXAMPLE OF GREEDY INITIALIZATION FOR MARL EXPLORATION

A greedy initialization can provide significant improvement in multi-agent exploration versus naïve random exploration, as shown by a simple thought experiment. Consider a two-player **MG** defined by a $4 \times 3$ gridworld with unit actions (up, down, left, right). Agent $A$ starts at (1,2) with goal (4,2), while agent $B$ starts at (4,2) with goal (1,2). The *greedy policy* for each agent in **MG** is to move horizontally toward its target, since this is optimal in the induced **M** (when the other agent is absent). Case 1: Suppose that for $\epsilon \in (0, 1)$, $A$ and $B$ follow greedy policies with probability $1 - \epsilon$, and take random actions ($p(a) = 1/4$) with probability $\epsilon$. Then the probability of a symmetric optimal trajectory is $P(\text{cooperate}) = 2\epsilon^2((1 - \epsilon) + \epsilon/4)^8$. For $\epsilon = 0.5$, $P(\text{cooperate}) \approx 0.01$. Case 2: If agents execute uniform random exploration, then $P(\text{cooperate}) = 3.05\text{e-}5 \ll 0.01$.

## E    GENERALIZATION

Table 1: Test performance with heavy traffic on difficult initial and goal lanes configurations

| Config | Initial lanes | Goal lanes | CM3 | IAC | COMA |
|--------|---------------|------------|-----|-----|------|
| C1 | $[1, 2]$ | $[3, 0]$ | **16.17** | 11.40 | 10.00 |
| C2 | Unif. random | Unif. random | **14.93** | 12.20 | 12.93 |
| C3 | $[1, 2]$ | $[2, 1]$ | **15.85** | 14.32 | 15.00 |
| C4 | $[0, 1]$ | $[3, 2]$ | **16.35** | 9.73 | 8.1 |

We investigated whether policies trained with few agent vehicles ($N = 2$) on an empty road can generalize to situations with heavy SUMO-controlled traffic. We also tested on initial and goal lane configurations (C3 and C4) which occur with low probability when training with configurations C1 and C2. Table 1 shows the sum of agents' reward, averaged over 100 test episodes, on these configurations that require cooperation with each other and with minimally-interactive SUMO-controlled vehicles for success. CM3's higher performance than IAC and COMA in training is reflected by better generalization performance on these test configurations. There is almost negligible decrase in performance from train Figure 5d to test, giving evidence to our hypothesis that centralized training with few agents is feasible even for deployment in situations with many agents, for certain applications where local interactions are dominant.

## F    ABSOLUTE RUNTIME

CM3's higher sample efficiency does not come at greater computational cost, as all methods' runtimes are within an order of magnitude of one another. Test times have no significant difference as all neural networks were similar.

Table 2: Absolute training runtime of all algorithms in seconds

| Environment | CM3 | IAC | COMA | QMIX |
|-------------|-----|-----|------|------|
| Antipodal | 1.1e4±348 | 0.9e4±20 | 1.9e4±238 | 1.0e4±19 |
| Cross | 1.9e4±256 | 1.5e4±26 | 1.3e4±12 | 1.1e4±34 |
| Merge | 8.5e3±21 | 6.8e3±105 | 9.6e3±294 | 1.2e4±61 |
| SUMO | 9.6e3±278 | 7.0e3±1.5e3 | 8.7e3±1.3e3 | 6.3e3±21 |
| Checkers | 9.2e3±880 | 8.5e3±568 | 7.7e3±2.2e3 | 11e3±1.4e3 |

## G    ENVIRONMENT DETAILS

The full Markov game for each experimental domain, along with the single-agent MDP induced from the Markov game, are defined in this section. In all domains, each agent's observation in the Markov game consists of two components, $o_{\text{self}}$ and $o_{\text{others}}$. CM3 leverages this decomposition for faster training, while IAC, COMA and QMIX do not.

### G.1 COOPERATIVE NAVIGATION

This domain is adapted from the multi-agent particle environment in Lowe *et al.* (2017). Movable agents and static landmarks are represented as circular objects located in a 2D unbounded world with real-valued position and velocity. Agents experience contact forces during collisions. A simple model of inertia and friction is involved.

**State.** The global state vector is the concatenation of all agents' absolute position $(x, y) \in \mathbb{R}^2$ and velocity $(v_x, v_y) \in \mathbb{R}^2$.

**Observation.** Each agent's observation of itself, $o_{\text{self}}$, is its own absolute position and velocity. Each agent's observation of others, $o_{\text{others}}$, is the concatenation of the relative positions and velocities of all other agents with respect to itself.

**Actions.** Agents take actions from the discrete set do nothing, up, down, left, right, where the movement actions produce an instantaneous velocity (with inertia effects).

**Goals and initial state assignment.** With probability 0.2, landmarks are given uniform random locations in the set $(-1, 1)^2$, and agents are assigned initial positions uniformly at random within the set $(-1, 1)^2$. With probability 0.8, they are predefined as follows (see Figure 3). In "Antipodal", landmarks for agents 1 to 4 have $(x, y)$ coordinates [(0.9,0.9), (-0.9,-0.9), (0.9,-0.9), (-0.9,0.9)], while agents 1 to 4 are placed at [(-0.9,-0.9), (0.9,0.9), (-0.9,0.9), (0.9,-0.9)]. In "Intersection", landmark coordinates are [(0.9,-0.15), (-0.9,0.15), (0.15,0.9), (-0.15,-0.9)], while agents are placed at [(-0.9,-0.15), (0.9,0.15), (0.15,-0.9), (-0.15,0.9)]. In "Merge", landmark coordinates are [(0.9,-0.2), (0.9,0.2)], while agents are [(-0.9,0.2), (-0.9,-0.2)]. Each agent's goal is the assigned landmark position vector.

**Reward.** At each time step, each agent's individual reward is the negative distance between its position and the position of its assigned landmark. If a collision occurs between any pair of agents, both agents receive an additional -1 penalty. A collision occurs when two agents' distance is less than the sum of their radius.

**Termination.** Episode terminates when all agents are less than 0.05 distance from assigned landmarks.

**Induced MDP.** This is the $N = 1$ case of the Markov game, used by Stage 1 of CM3. The single agent only receives $o_{\text{self}}$. In each episode, its initial position and the assigned landmark's initial position are both uniform randomly chosen from $(-1, 1)^2$.

### G.2 SUMO

We constructed a straight road of total length 200m and width 12.8m, consisting of four lanes. All lanes have width 3.2m, and vehicles can be aligned along any of four sub-lanes within a lane, with lateral spacing $0.8m$. Vehicles are emitted at average speed 30m/s with small deviation. Simulation time resolution was $0.2s$ per step. SUMO file `merge_stage3_dense.rou.xml` contains all vehicle parameters, and `merge.net.xml` defines the complete road architecture.

**State.** The global state vector $s$ is the concatenation of all agents' absolute position $(x, y)$, normalized respectively by the total length and width of the road, and horizontal speed $v$ normalized by 29m/s.

**Observation.** Each agent observation of itself $o_{\text{self}}^n$ is a vector consisting of: agent speed normalized by 29m/s, normalized number of sub-lanes between agent's current sub-lane and center sub-lane of goal lane, and normalized longitudinal distance to goal position. Each agent's observation of others $o_{\text{others}}^n$ is a discretized observation tensor of shape [13,9,2] centered on the agent, with two channels: binary indicator of vehicle occupancy, and normalized relative speed between agent and other vehicles. Each channel is a matrix with shape [13,9], corresponding to visibility of $15m$ forward and backward (with resolution $2.5m$) and four sub-lanes to the left and right.

**Actions.** All agents have the same discrete action space, consisting of five options: no-op (maintain current speed and lane), accelerate ($2.5m/s^2$), decelerate ($-2.5m/s^2$), shift one sub-lane to the left, shift one sub-lane to the right. Each agent's action $a^n$ is represented as a one-hot vector of length 5.

**Goals and initial state assignment.** Each goal vector $g^n$ is a one-hot vector of length 4, indicating the goal lane at which agent $n$ should arrive once it crosses position $x$=190m. With probability 0.2, agents are assigned goals uniformly at random, and agents are assigned initial lanes uniformly at

random at position $x$=0. With probability 0.8, agent 1's goal is lane 2 and agent 2's goal is lane 1, while agent 1 is initialized at lane 1 and agent 2 is initialized at lane 2 (see Figure 4). Departure times were drawn from a normal distribution with mean 0s and standard deviation 0.5s for each agent.

**Reward.** The reward $R(s_t, \mathbf{a}_t, g^n)$ for agent $n$ with goal $g^n$ is given according to the conditions: -1 for a collision; -10 for time-out (exceed 33 simulation steps during an episode); $10(1 - \Delta)$ for reaching the end of the road and having a normalized sub-lane difference of $\Delta$ from the center of the goal lane; and -0.1 if current speed exceeds 35.7m/s.

**Termination.** Episode terminates when 33 simulation steps have elapsed or all agents have $x >$190m.

**Induced MDP.** This is the $N = 1$ case of the Markov game defined above, used by Stage 1 of CM3. The single agent receives only $o_{\text{self}}$. For each episode, agent initial and goal lanes are assigned uniformly at random from the available lanes.

### G.3 CHECKERS

This domain is adapted from the Checkers environment in Sunehag *et al.* (2018). It is a gridworld with 5 rows and 13 columns (Figure 2). Agents cannot move to the two highest and lowest rows and the two highest and lowest columns, which are placed for agents' finite observation grid to be well-defined. Agents cannot be in the same grid location. Red and yellow collectible reward are placed in a checkered pattern in the middle 3x8 region, and they disappear when any agent moves to their location.

**State.** The global state $s$ consists of two components. The first is $s_T$, a tensor of shape [3,9,2], where the two "channels" in the last dimension represents the presence/absence of red and yellow rewards as 1-hot matrices. The second is $s_V$, the concatenation of all agents' $(x, y)$ location (integer-valued) and the number of red and yellow each agent has collected so far.

**Observation.** Each agent's obsevation of others, $o_{\text{others}}^n$, is the concatenation of all other agents' normalized coordinates (normalized by total size of grid). An agent's observation of itself, $o_{\text{self}}^n$, consists of two components. First, $o_{\text{self},V}^n$ is a vector concatenation of agent $n$'s normalized coordinate and the number of red and yellow it has collected so far. Second, $o_{\text{self},T}^n$ is a tensor of shape [5,5,3], centered on its current location in the grid. The tensor has three "channels", where the first two represent presence/absence of red and yellow rewards as 1-hot matrices, and the last channel indicates the invalid locations as a 1-hot matrix. The agent's own grid location is a valid location, while other agents' locations are invalid.

**Actions.** Agents choose from a discrete set of actions do-nothing, up, down, left, right. Movement actions transport the agent one grid cell in the chosen direction.

**Goals.** Agent A's goal is to collect all red rewards without touching yellow. Agent B's goal is to collect all yellow without touching red. The goal is represented as a 1-hot vector of length 2.

**Reward.** Agent A gets +1 for red, -0.5 for yellow. Agent B gets -0.5 for red, +1 for yellow.

**Initial state distribution.** Agent A is initialized at (2,8), Agent B is initialized at (4,8). (0,0) is the top-left cell (Figure 2).

**Termination.** Each episode finishes when either 75 time steps have elapsed, or when all rewards have been collected.

**Induced MDP.** For Stage 1 of CM3, the single agent is randomly assigned the role of either Agent A or Agent B in each episode. Everything else is defined as above.

## H ARCHITECTURE

For all experiment domains, ReLU nonlinearity was used for all neural network layers unless otherwise specified. All layers are fully-connected feedforward layers, unless otherwise specified. All experiment domains have a discrete action space (with $|\mathcal{A}| = 5$ actions), and action probabilities were computed by lower-bounding softmax outputs of all policy networks by $P(a^n = i) = (1 - \epsilon)\text{softmax}(i) + \epsilon/|\mathcal{A}|$, where $\epsilon$ is a decaying exploration parameter. To keep neural network architectures as similar as possible among all algorithms, our neural networks for

COMA differ from those of Foerster *et al.* (2018) in that we do not use recurrent networks, and we do not feed previous actions into the Q function. For the Q network in all implementations of COMA, the value of each output node $i$ is interpreted as the action-value $Q(s, a^{-n}, a^n = i, \mathbf{g})$ for agent $n$ taking action $i$ and all other agents taking action $a^{-n}$. Also for COMA, agent $n$'s label vector (one-hot indicator vector) and observation $o_{\text{self}}$ were used as input to COMA's global Q function, to differentiate between evaluations of the Q-function for different agents. These were choices in Foerster *et al.* (2018) that we retain.

## H.1 COOPERATIVE NAVIGATION

**CM3.** The policy network $\pi^1$ in Stage 1 feeds the concatenation of $o_{\text{self}}$ and goal $g$ to one layer with 64 units, which is connected to the special layer $h^1_*$ with 64 units, then connected to the softmax output layer with 5 units, each corresponding to one discrete action. In Stage 2, $o_{\text{others}}$ is connected to a new layer with 128 units, then connected to $h^1_*$.

The $Q^1$ function in Stage 1 feeds the concatenation of state $s$, goal $g$, and 1-hot action $a$ to one layer with 64 units, which is connected to the special layer $h^1_*$ with 64 units, then to a single linear output unit. In Stage 2, $Q^1$ is augmented into both $Q^{\boldsymbol{\pi}}_n(s, \mathbf{a})$ and $Q^{\boldsymbol{\pi}}_n(s, a^m)$ as separate networks. For $Q^{\boldsymbol{\pi}}_n(s, \mathbf{a})$, $s^{-n}$ (part of state $s$ excluding agent $n$) and $a^{-n}$ are concatenated and connected to a layer with 128 units, then connected to $h^1_*$. For $Q^{\boldsymbol{\pi}}_n(s, a^m)$, $s^m$ (agent $m$ portion of state $s$) and $s^{-n}$ are concatenated and connected to a layer with 128 units, then connected to $h^1_*$.

**IAC.** IAC uses the same policy network as Stage 2 of CM3. The value function of IAC concatenates $o^n_{\text{self}}$ and goal $g^n$, connects to a layer with 64 units, which connects to a second layer $h_2$ with 64 units, then to a single linear output unit. $o^n_{\text{others}}$ is connected to a layer with 128 units, then connected to $h_2$.

**COMA.** COMA uses the same policy network as Stage 2 of CM3. The global Q function of COMA computes $Q(s, (a^n, a^{-n}))$ for each agent $n$ as follows. Input is the concatenation of state $s$, all other agents' 1-hot actions $a^{-n}$, agent n's goal $g^n$, all other agent goals $g^{-n}$, agent label $n$, and agent $n$'s observation $o^n_{\text{self}}$. This is passed through two layers of 128 units each, then connected to a linear output layer with 5 units.

**QMIX.** Individual value functions take input $(o^n_{\text{self}}, o^n_{\text{others}}, g^n)$ and connects to one hidden layer with 64 units, which connects to the output layer. The mixing network follows the exact architecture of Rashid *et al.* (2018) with embedding dimension 64.

## H.2 SUMO

**CM3.** The policy network $\pi^1$ during Stage 1 feeds each of the inputs $o_{\text{self}}$ and goal $g^n$ to a layer with 32 units. The concatenation is then connected to the layer $h^1_*$ with 64 units, and connected to a softmax output layer with 5 units, each corresponding to one discrete action. In Stage 2, the input observation grid $o^n_{\text{others}}$ is processed by a convolutional layer with 4 filters of size 5x3 and stride 1x1, flattened and connected to a layer with 64 units, then connected to the layer $h^1_*$ of $\pi^1$.

The $Q^1$ function in Stage 1 feeds the concatenation of state $s$, goal $g$, and 1-hot action $a$ to one layer with 256 units, which is connected to the special layer $h^1_*$ with 256 units, then to a single linear output unit. In Stage 2, $Q^1$ is augmented into both $Q^{\boldsymbol{\pi}}_n(s, \mathbf{a})$ and $Q^{\boldsymbol{\pi}}_n(s, a^m)$ as separate networks. For $Q^{\boldsymbol{\pi}}_n(s, \mathbf{a})$, $s^{-n}$ (part of state $s$ excluding agent $n$), $a^{-n}$, and $g^{-n}$ are concatenated and connected to a layer with 128 units, then connected to $h^1_*$. For $Q^{\boldsymbol{\pi}}_n(s, a^m)$, $s^m$ (agent $m$ portion of state $s$), $s^{-n}$, and $g^{-n}$ are concatenated and connected to a layer with 128 units, then connected to $h^1_*$.

**IAC.** IAC uses the same policy network as Stage 2 of CM3. The value function of IAC concatenates $o^n_{\text{self}}$ and $g^n$, feeds it into a layer with 64 units, which connects to a layer $h_2$ with 64 units, which connects to one linear output unit. $o^n_{\text{others}}$ is processed by a convolutional layer with 4 filters of size 5x3 and stride 1x1, flattened and connected to a layer with 128 units, then connected to $h_2$.

**COMA.** COMA uses the same policy network as Stage 2 of CM3. The Q function of COMA is exactly the same as the one in COMA for cooperative navigation defined above.

**QMIX.** Individual value functions take input $(o^n_{\text{self}}, g^n)$ and connects to one hidden layer with 64 units, which connects to layer $h_2$ with 64 units. $o^n_{\text{others}}$ is passed through the same convolutional layer

as above and connected to $h_2$. $h_2$ is fully-connected to an output layer. The mixing network follows the exact architecture of Rashid *et al.* (2018) with embedding dimension 64.

### H.3 CHECKERS

**CM3.** The policy network $\pi^1$ during Stage 1 feeds $o^n_{\text{self},T}$ to a convolution layer with 6 filters of size 3x3 and stride 1x1, which is flattened and connected to a layer with 32 units, which is concatenated with $o^n_{\text{self},V}$, previous action, and its goal vector. The concatenation is connected to a layer with 256 units, then to the special layer $h^1_*$ with 256 units, finally to a softmax output layer with 5 units. In Stage 2, $o^n_{\text{others}}$ is connected to a layer with 256 units, then to the layer $h^1_*$ of $\pi^1$.

The $Q^1$ function in Stage 1 is defined as: state tensor $s_T$ is fed to a convolutional layer with 4 filters of size 3x5 and stride 1x1 and flattened. $o^n_{\text{self},T}$ is given to a convolution layer with 6 filters of size 3x3 and stride 1x1 and flattened. Both are concatenated with $s^n$ (agent $n$ part of the $s_V$ vector), goal $g^n$, action $a^n$ and $o^n_{\text{self},V}$. The concatenation is fed to a layer with 256 units, then to the special layer $h^1_*$ with 256 units, then to a single linear output unit. In Stage 2, $Q^1$ is augmented into both $Q^{\boldsymbol{\pi}}_n(s, \mathbf{a})$ and $Q^{\boldsymbol{\pi}}_n(s, a^m)$ as separate networks. For $Q^{\boldsymbol{\pi}}_n(s, \mathbf{a})$, $s^{-n}$ (part of state vector $s_V$ excluding agent $n$) and $a^{-n}$ are concatenated and connected to a layer with 32 units, then connected to $h^1_*$. For $Q^{\boldsymbol{\pi}}_n(s, a^m)$, $s^m$ (agent $m$ portion of state $s_V$) and $s^{-n}$ are concatenated and connected to a layer with 32 units, then connected to $h^1_*$.

**IAC.** IAC uses the same policy network as Stage 2 of CM3. The value function of IAC feeds $o^n_{\text{self},T}$ to a convolutional layer with 6 filters of size 3x3 and stride 1x1, which is flattened and concatenated with $o^n_{\text{self},V}$ and goal $g^n$. The concatenation is connected to a layer with 256 units, then to a layer $h_2$ with 256 units, then to a single linear output unit. $o^n_{\text{others}}$ is connected to a layer with 32 units, then to the layer $h_2$.

**COMA.** COMA uses the same policy network as Stage 2 of CM3. The global $Q(s, (a^n, a^{-n}))$ function of COMA is defined as follows for each agent $n$. Tensor part of global state $s_T$ is given to a convolutional layer with 4 filters of size 3x5 and stride 1x1. Tensor part of agent $n$'s observation $o^n_{\text{self},T}$ is given to a convolutional layer with 6 filters of size 3x3 and stride 1x1. Outputs of both convolutional layers are flattened, then concatenated with $s_V$, all other agents' actions $a^{-n}$, agent $n$'s goal $g^n$, other agents' goals $g^{-n}$, agent $n$'s label vector, and agent $n$'s vector observation $o^n_{\text{self},V}$. The concatenation is passed through two layers with 256 units each, then to a linear output layer with 5 units.

**QMIX.** Individual value functions are defined as: $o^n_{\text{self},T}$ is passed through the same convolutional layer as above, connected to hidden layer with 32 units, then concatenated with $o^n_{\text{self},V}$, $a^n_{t-1}$, and $g^n$. This is connected to layer $h_2$ with 64 units. $o^n_{\text{others}}$ is connected to a layer with 64 units then connectd to $h_2$. $h_2$ is fully-connected to an output layer. The mixing network feeds $s_T$ into the same convolutional network as above and follows the exact architecture of Rashid *et al.* (2018) with embedding dimension 128.

## I PARAMETERS

We used the Adam optimizer in Tensorflow with hyperparameters in Tables 3 to 5. $\epsilon_{\text{div}}$ is used to compute the exploration decrement $\epsilon_{\text{step}} := (\epsilon_{\text{start}} - \epsilon_{\text{end}})/\epsilon_{\text{div}}$.

Table 3: Parameters used for CM3, ablations, and baselines in cooperative navigation

| Parameter | CM3 | | | | IAC | COMA | QMIX |
|---|---|---|---|---|---|---|---|
| | Stage 1 | Stage 2 | QV | Direct | | | |
| Episodes | 1e3 | 8e4 | 8e4 | 8e4 | 8e4 | 8e4 | 8e4 |
| $\epsilon_{start}$ | 1.0 | 0.5 | 0.5 | 1.0 | 1.0 | 1.0 | 1.0 |
| $\epsilon_{end}$ | 0.01 | 0.05 | 0.05 | 0.05 | 0.05 | 0.05 | 0.05 |
| $\epsilon_{div}$ | 1e3 | 2e4 | 2e4 | 8e4 | 8e4 | 2e4 | 8e4 |
| Replay buffer | 1e4 | 1e4 | 1e4 | 1e4 | 1e4 | 1e4 | 1e4 |
| Minibatch size | 256 | 128 | 128 | 128 | 128 | 128 | 128 |
| Episodes per train | 10 | 10 | 10 | 10 | 10 | 10 | N/A |
| Learning rate $\pi$ | 1e-4 | 1e-4 | 1e-4 | 1e-4 | 1e-4 | 1e-5 | N/A |
| Learning rate $Q$ | 1e-3 | 1e-3 | 1e-3 | 1e-3 | N/A | 1e-4 | 1e-3 |
| Learning rate $V$ | N/A | N/A | 1e-3 | N/A | 1e-3 | N/A | N/A |
| Epochs | 24 | 24 | 24 | 24 | 24 | 24 | NA |
| Steps per train | N/A | N/A | N/A | N/A | N/A | N/A | 10 |
| Max env steps | 25 | 50 | 50 | 50 | 50 | 50 | 50 |

Table 4: Parameters used for CM3 and baselines in SUMO

| Parameter | CM3 | | | | IAC | COMA | QMIX |
|---|---|---|---|---|---|---|---|
| | Stage 1 | Stage 2 | QV | Direct | | | |
| Episodes | 2.5e3 | 5e4 | 5e4 | 5e4 | 5e4 | 5e4 | 5e4 |
| $\epsilon_{start}$ | 0.5 | 0.5 | 0.5 | 0.5 | 0.5 | 0.5 | 0.5 |
| $\epsilon_{end}$ | 0.05 | 0.05 | 0.05 | 0.05 | 0.05 | 0.05 | 0.05 |
| $\epsilon_{step}$ | 2e3 | 1e3 | 4e4 | 4e4 | 1e3 | 1e4 | 4e4 |
| Replay buffer | 1e4 | 2e4 | 2e4 | 2e4 | 2e4 | 2e4 | 2e4 |
| Minibatch size | 128 | 128 | 128 | 128 | 128 | 128 | 128 |
| Steps per train | 10 | 10 | 10 | 10 | N/A | N/A | 10 |
| Episodes per train | N/A | N/A | N/A | N/A | 10 | 10 | N/A |
| Learning rate $\pi$ | 1e-4 | 1e-4 | 1e-4 | 1e-4 | 1e-4 | 1e-4 | N/A |
| Learning rate $Q$ | 1e-3 | 1e-3 | 1e-3 | 1e-3 | N/A | 1e-3 | 1e-3 |
| Learning rate $V$ | N/A | N/A | 1e-3 | N/A | 1e-3 | N/A | N/A |
| Epochs | N/A | N/A | N/A | N/A | 33 | 33 | N/A |
| Max env steps | 33 | 33 | 33 | 33 | 33 | 33 | 33 |

Table 5: Parameters used for CM3 and baselines in Checkers

| Parameter | CM3 | | | | IAC | COMA | QMIX |
|---|---|---|---|---|---|---|---|
| | Stage 1 | Stage 2 | QV | Direct | | | |
| Episodes | 5e3 | 5e4 | 5e4 | 5e4 | 5e4 | 5e4 | 5e4 |
| $\epsilon_{start}$ | 1.0 | 0.5 | 0.5 | 1.0 | 1.0 | 1.0 | 1.0 |
| $\epsilon_{end}$ | 0.1 | 0.1 | 0.1 | 0.1 | 0.1 | 0.1 | 0.1 |
| $\epsilon_{step}$ | 5e2 | 1e3 | 1e3 | 1e4 | 2e4 | 1e4 | 1e4 |
| Replay buffer | 1e4 | 1e4 | 1e4 | 1e4 | 1e4 | 1e4 | 1e4 |
| Minibatch size | 128 | 128 | 128 | 128 | 128 | 128 | 128 |
| Steps per train | N/A | 10 | 10 | 10 | N/A | N/A | 10 |
| Episodes per train | 10 | N/A | N/A | N/A | 10 | 10 | N/A |
| Learning rate $\pi$ | 1e-4 | 1e-4 | 1e-4 | 1e-4 | 1e-4 | 1e-4 | N/A |
| Learning rate $Q$ | 1e-3 | 1e-3 | 1e-3 | 1e-3 | N/A | 1e-3 | 1e-5 |
| Learning rate $V$ | N/A | N/A | 1e-3 | N/A | 1e-3 | N/A | N/A |
| Epochs | 10 | N/A | N/A | N/A | 33 | 33 | N/A |
| Max env steps | 75 | 75 | 75 | 75 | 75 | 75 | 75 |

## J  STAGE 1

The Stage 1 functions $Q^1$ and $\pi^1$ for a single agent are trained with the $N = 1$ equivalents of (4) and (5):

$$L(\theta_Q) = \mathbb{E}_{\boldsymbol{\pi}}\left[\left(y_i - Q^1_{\theta_Q}(s_i, a_i)\right)^2\right] \tag{14}$$

$$y_i := R(s_i, \mathbf{a}_i, g^n) + \gamma Q^1_{\theta_Q}(s_{i+1}, a_{i+1}) \tag{15}$$

$$\nabla_\theta J(\pi^1) = \mathbb{E}_{\pi^1}\left[\nabla_\theta \log \pi(a)\left(Q^{\pi^1}(s, a) - \sum_{\hat{a}} \pi^1(\hat{a})Q^{\pi^1}(s, \hat{a})\right)\right] \tag{16}$$

Stage 1 training curves for all three experimental domains are shown in Figure 6.

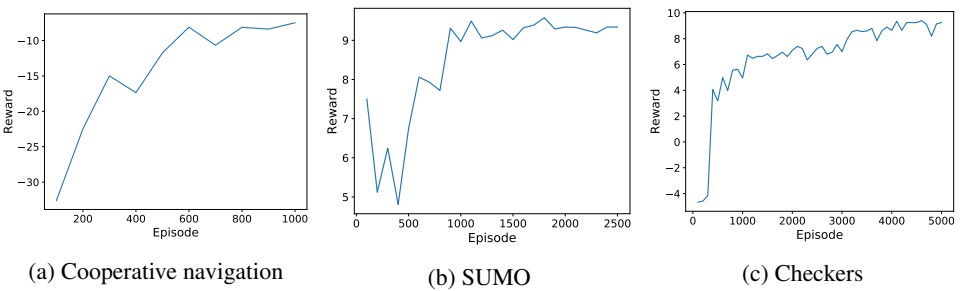

(a) Cooperative navigation    (b) SUMO    (c) Checkers

Figure 6: Stage 1 reward curves for CM3 in cooperative navigation, SUMO and Checkers.

