# OpenReview forum: "CM3: Cooperative Multi-goal Multi-stage Multi-agent Reinforcement Learning"
_ICLR.cc/2020/Conference — Accept (Poster)_

### Official Review · AnonReviewer2 · 2019-10-23
**Official Blind Review #2**

**Rating:** 6

**Review:**

Contribution:

This paper derives a loss for cooperative MARL for the case where agents have individual goals assigned to them.
The assumption of individual goals allows for a better credit assignment, as well as effective pre-training of one agent to solve its own goal in isolation. Based on this observation, the paper subsequently proposes a curriculum learning scheme to fully take advantage of this property.
Experiments in varied domains are shown.

Review:

The paper is well written and easy to follow, and makes a good case by having a thorough experimental analysis, as well as a theoretical analysis of the credit function.

The applicability of the method seems rather high, even though there exists multiple MARL environments where the multi-goal assumption will be broken. In a predator-prey domain, for example, the predators can't learn anything because their tasks is not solvable with only one agent. In that context, it seems that stage 1 would be useless, but would stage 2 still work, and if yes how would that compare to other baseline methods?

Some details of the multi-stage training are a bit unclear to me. Section 4.5 states "we train an actor \pi^1 and critic Q^1 to convergence [...]". However, I don't fully grasp how this agent will be able to learn to solve all the goals? It seems to me that with N=1, equation (5) reduces to normal actor critic with one agent and one goal, but I'd expect that the policy must be trained on all the goals, as this is hinted at in section 5 (" in Checkers, we alternate between training one agent as A and B"). Could you clarify that part?

About the function augmentation, could you clarify how the new network \pi^2 is initialized? It seems that the initial values of W^1 and W^{1:2} in particular are quite important, because if the resulting policy is too far off from the initial policy \pi^1, then the benefit of the pre-training could be lost on the way. Did you find that any special care like initializing W1 = I and W^{1:2} = 0 is required here?


One minor complaint is that the proposed method never seems to achieve statistically better performance than the best baseline on any of the tasks (for cooperative navigation it is tied with IAC and for SUMO and Checkers it is tied with QMIX). But since the best baseline is different across tasks, it suggests that the proposed method is more versatile.


A potentially relevant missed reference: [1].


[1] A Structured Prediction Approach for Generalization in Cooperative Multi-Agent Reinforcement Learning, Carion et al, https://arxiv.org/abs/1910.08809


**Experience Assessment:**

I have published one or two papers in this area.

**Review Assessment: Checking Correctness Of Derivations And Theory:**

I assessed the sensibility of the derivations and theory.

**Review Assessment: Checking Correctness Of Experiments:**

I carefully checked the experiments.

**Review Assessment: Thoroughness In Paper Reading:**

I read the paper at least twice and used my best judgement in assessing the paper.

---

> ### Author Response · Authors · 2019-11-12
> **We appreciate Reviewer 2's thorough reading, constructive review, and reference to complementary work**
>
> We thank Reviewer 2 for the positive feedback on the motivation, analysis and experimental results of the paper, and for raising pertinent and constructive questions.
>
> Regarding applicability of multi-goal MARL:
> There are two cases to consider. 1) The problem cannot be formulated as cooperative multi-goal MARL, because there is a single team goal that cannot be decomposed into individual goals in a meaningful way. A predator-prey game may fall under this case (depending on specifics), if all predators share the same goal of capturing one prey and receive a team reward for doing so. While we acknowledge there are interesting MARL settings outside of multi-goal MARL (e.g., pure communication games, as we mentioned in the text), we also provided many important real-world problems that are naturally modeled as multi-goal MARL, e.g., autonomous driving, traffic light control, warehouse commissioning, and multi-player games among others.
> 2) The multi-goal MARL formulation applies, but appears to pose difficulties for CM3’s curriculum approach. In fact, the Checkers game we used for experiments falls under this case: as we described in Section 5, it is impossible for Agent A to get the maximum possible rewards if Agent B is not available to help it clear the squares with penalties for Agent A, and vice versa for Agent B. However, Figure 5(e,j) shows that the CM3 curriculum still helped, because the single agent learned useful behavior to attain non-zero rewards in the induced MDP. This suggests that as long as a single agent can receive some rewards in the induced MDP, CM3’s curriculum can be applied.
> Training only Stage 2 is equivalent to the ablation that we call 'Direct' in Figure 5. Antipodal, Cross, and SUMO show that Stage 1 is important for CM3, while results on Merge and Checkers show that CM3 can be competitive with baselines in some cases even with Stage 1 removed.
>
> Regarding convergence or reaching all goals in Stage 1:
> Reviewer 2 is correct in stating that the actor-critic in Stage 1 is trained on all goals, up to random sampling. When N=1, equation (5) reduces to a single-agent actor critic, but with the key difference that the assigned goal g is randomly sampled from the set G in each episode. As stated in Section 4.5, the sampling of goal(s) is done each episode, for both the single-agent Stage 1 and the multi-agent Stage 2.
>
> Regarding initialization for the Stage 2 networks:
> To clarify, W^1 refers to all the weights that are trained in Stage 1, so they are carried over to Stage 2 (i.e., restored from a model checkpoint), and they are not re-initialized at the start of Stage 2. We initialized the connection weights W^{1:2} with a truncated Normal(0, 0.01), matching the intuition that they should have small initial impact. Furthermore, the overall network outputs are still taken from the original outputs of \pi^1, and only a single layer h^1_{i*} is directly affected by W^{1:2}, which mitigates the impact of initialization of Stage 2.
>
> Regarding performance versus baselines:
> The fact that CM3 and a baseline saturate at the same maximum performance indicates there is room to experiment on more complex tasks. We appreciate that Reviewer 2 noticed the best baseline is different across tasks, while CM3 is robust on all three tasks. This motivates future investigation into whether and why different environments pose qualitatively different challenges for different algorithms. The ablations we conducted on CM3 is a start to understand which techniques are needed in which tasks. On the other hand, training speed is also a key metric in MARL, and here we do see statistically significant speedup of CM3 over all baselines except versus QMIX on SUMO.
>
> We agree that Carion et al. 2019 is a relevant work that is complementary to ours, as they focus on optimizing high-level agent-task assignment given fixed low-level policies, while we focus on curriculum and credit assignment for low-level policies given that each agent is assigned a goal. We have updated the paper to reflect this.
>
> Again, we thank Reviewer 2 for constructive feedback.

---

### Official Review · AnonReviewer3 · 2019-10-24
**Official Blind Review #3**

**Rating:** 6

**Review:**

The paper presents a new method CM3 for multi-goal multi-agent settings where agent must care about the rewards of others as well as theirs. The method restructures the problem into two-stage curriculum where individual agents solve the problem in the individual setting of the environment in the first stage and then more agents are introduced in the multi-agent setting in second stage. The method also uses function augmentation to only learn parameters which are necessary for single agent in first stage and then more parameters are introduced in the second stage. The results are shown on three environments which show that CM3 outperforms the baselines.

The paper is well-written, motivated and clear to read. A lot of important stuff has been pushed into appendix and experiments/setup require more details but overall I believe paper has significant contributions and results. Therefore, I assign a rating of weak accept which I am happy to raise if clarity of paper can be improved.

The paper doesn’t comment on parameter counts of CM3 compared with baselines which is also an important factor in choosing one method over the other. I would like to see more quantitative analysis as presented in IAC to understand what is happening behind the scenes. It is very surprising to see that IAC is outperforming COMA in most of the tasks which is opposite of what COMA paper suggested.

On the note of curriculum [1] and [2] uses curriculum in traffic junction settings to improve overall performance as agent increases. This can be compared to moving from stage 1 to stage 2 but instead we move from less number of agents to more. [2] also suggests that using individualized rewards help in better credit assignment. There is also an assumption in the setup that private observations from all other agents are readily available. It would be interesting to see what would happen if agents have to communicate their private state. So, the paper is also missing discussion on communication protocols (discrete, continuous, through critic) [1][2].

It is hard to directly compare through the charts. So, tables in Appendix E and F should be moved to the main text. The experiment section needs to be extended and the main model section needs to be decreased in the content amount.

It seems like you missed 1/n in advantage function’s equation in Section 3, Multi-agent credit assignment section. Figure 6 has been mentioned in 6 second paragraph but doesn’t appear until the last page.

[1] Sukhbaatar, Sainbayar, and Rob Fergus. "Learning multiagent communication with backpropagation." In Advances in Neural Information Processing Systems, pp. 2244-2252. 2016.
[2] Singh, Amanpreet, Tushar Jain, and Sainbayar Sukhbaatar. "Learning when to communicate at scale in multiagent cooperative and competitive tasks." arXiv preprint arXiv:1812.09755 (2018).

**Experience Assessment:**

I have published one or two papers in this area.

**Review Assessment: Checking Correctness Of Derivations And Theory:**

I assessed the sensibility of the derivations and theory.

**Review Assessment: Checking Correctness Of Experiments:**

I assessed the sensibility of the experiments.

**Review Assessment: Thoroughness In Paper Reading:**

I read the paper at least twice and used my best judgement in assessing the paper.

---

> ### Author Response · Authors · 2019-11-12
> **We appreciate the constructive feedback and have updated the paper to discuss the relevant references.**
>
> We thank Reviewer 3 for an accurate understanding of the work and for constructive feedback.
>
> We placed the detailed definition of state, observation, actions, goals, reward and induced MDP for the 3 multi-goal Markov games in Appendix G because we want to give a full written description for reproducibility to accompany our code to be released. We also give a full written description of all neural networks for all methods in Appendix H. Combined, they take up 4-5 pages. We will continue to make an effort to move some of these details into the main text.
>
> Regarding the number of trainable parameters, we state in Algorithm Implementations in Section 5 that precise architecture details are in Appendix H. Except for QMIX’s hypernetwork, all other neural networks have up to two hidden layers and one output layer, with convnets for processing the image part of observation and state in SUMO and Checkers. Regarding hyperparameters, CM3 has the same core set of hyperparameters as the baselines (see Appendix I). Analogous to QMIX’ use of a hypernetwork, the function augmentation in CM3 is an additional degree of freedom but we did not face the need to tune the choice of layer i* in our experiments.
>
> Regarding IAC outperforming COMA and QMIX in cooperative navigation:
> The updated version includes additional explanation. As Reviewer 3 point out, Singh et al. 2018 find that individual rewards help to resolve credit assignment. In Foerster et al. 2018, IAC learned from the global team reward. In our multi-goal setting, each IAC agent truly is independent and learns its own goal using its individual reward. Hence our experimental finding is consistent with Singh et al. 2018. We also note that the comparison of COMA to IAC in Foerster 2018 was conducted on the task of SC2 micromanagement, which may exhibit different challenges from the three tasks evaluated in our work. Nonetheless, our experiments in SUMO and Checkers confirm that IAC underperforms all other methods when the problem requires more sustained cooperative behavior.
>
> We agree that Sukhbaatar et al. 2016 and Singh et al. 2018 are relevant literature that provide empirical support for the benefits of curriculum learning and individual rewards in MARL. We include this in Section 2 and 6 in the latest version. The main difference is that we explicitly train Stage 1 for all goals, and we evaluate on tasks where cooperation require physical and strategic movement without access to communication.
>
> We work in the framework of centralized training with decentralized execution. We have added clarification in Section 4.4 that each individual agent does not use the private observation of any other agent as input to its policy. This is how we ensure decentralized execution. Instead, the centralized part of CM3 is the use of the credit function and the joint action-value function, both of which use the global state as input.
>
> Appendix E is a supplementary test of generalization in SUMO, not our primary metric of task performance and learning speed that we show in Figure 5. Appendix F describes the wall-clock time, which heavily depends on computational resources, and which is not a standard metric in the MARL literature (Hernandez-Leal et al. 2018). Hence we report the full training curve to show sample efficiency. We will improve the readability of Figure 5.
>
> The equation in “Multi-agent credit assignment” in Section 3 is Equation (4) in Foerster et al. 2018, which does not appear to have a 1/n factor. Figure 6 is supplementary material to show convergence within the allotted sample count for Stage 1. We only need to state the numerical sample count in Section 6 to compare sample efficiency, so we leave the full figure in the Appendix.
>
> We again express appreciation for Reviewer 3’s constructive feedback.

---

### Official Review · AnonReviewer1 · 2019-10-25
**Official Blind Review #1**

**Rating:** 6

**Review:**

This paper presents a method for adding credit assignment
to multi-agent RL and proposes a way of adding a curriculum
to the training process.
The best heuristics and structures to incorporate in the
modeling, learning, and exploration parts of multi-agent
RL are still largely unknown and this paper explores some
reasonable new ones.
In most of the tasks this is evaluated on in Figure 5
this approach adds a slight improvement to the SOTA
and I think an exciting direction of future work is
to continue pushing on multi-agent RL in even more
complex envirnoments where the SOTA break down in
even worse ways.


**Experience Assessment:**

I have read many papers in this area.

**Review Assessment: Checking Correctness Of Derivations And Theory:**

I did not assess the derivations or theory.

**Review Assessment: Checking Correctness Of Experiments:**

I assessed the sensibility of the experiments.

**Review Assessment: Thoroughness In Paper Reading:**

I made a quick assessment of this paper.

---

> ### Author Response · Authors · 2019-11-12
> **We thank Reviewer 1 for the encouraging assessment.**
>
> We thank Reviewer 1 for assessing our work, and we agree that investigating how best to introduce additional structure in MARL is a fruitful research direction, analogous to the case for single-agent RL. In addition to environment complexity, we believe that algorithms for MARL should be evaluated on diverse environment as well. As our experiments show, certain existing methods perform well in certain environments but underperform in others. Diversity of environments is needed to understand the relative strengths and weaknesses of each algorithm. The additional benefit of using a modular algorithm design, such as in CM3, is that ablation studies can give experimental guidance for the kinds of heuristics or structure that are useful for different kinds of problems and deserve more theoretical investigation.

---

> > ### Comment · AnonReviewer1 · 2019-11-15
> > **Response**
> >
> > Thanks for the responses. I've read through the other reviews and comments posted for this paper and still keep my original score of a weak accept.

---

### Decision · Program_Chairs · 2019-12-19

**Decision:**

Accept (Poster)

**Comment:**

This paper was generally well received by reviewers and was rated as a weak accept by all.
The AC recommends acceptance.